# Effect of Diallyl Trisulfide on the Reproductive Behavior of the Grain Moth, *Sitotroga cerealella* (Lepidoptera: Gelechiidae)

**DOI:** 10.3390/insects11010021

**Published:** 2019-12-25

**Authors:** Meng-Meng Chang, Sakhawat Shah, Meng-Ya Wu, Su-Su Zhang, Gang Wu, Feng-Lian Yang

**Affiliations:** Hubei Insect Resources Utilization and Sustainable Pest Management Key Laboratory, College of Plant Science and Technology, Huazhong Agricultural University, Wuhan 430070, China; changmengmeng@ioz.ac.cn (M.-M.C.); shahentomology@webmail.hzau.edu.cn (S.S.); wumengya@webmail.hzau.edu.cn (M.-Y.W.); zhangsusu@webmail.hzau.edu.cn (S.-S.Z.); wugang@mail.hzau.edu.cn (G.W.)

**Keywords:** botanicals, circadian mating rhythms, calling periodicity, garlic essential oil, inhibiting oviposition, pest management, stored product protection

## Abstract

The Angoumois grain moth, *Sitotroga cerealella* (Olivier, 1789) (Lepidoptera: Gelechiidae), is primarily a pest of stored products, that feeds inside the grain as larvae inducing significant economic loss in various stored commodities. Our previous studies proved that garlic essential oil and its active substances inhibit oviposition in moths. To further explore the effect on reproductive behavior and accurately interpret the results in terms of effective control of the moth population, moths were treated with diallyl trisulfide (DATS), an active substance of garlic essential oil, at a dose of 0.015 µL/L in air (LC_20_, sub-lethal concentration). The results showed that fecundity and the proportion of viable eggs significantly decreased when the moths were treated with LC_20_ DATS. Furthermore, female circadian mating rhythms and calling periodicity changed significantly after treatment. Compared with controls, the peak in mating occurred approximately 1 h earlier on the first day after DATS treatment, while it was significantly later on days 2 and 3. Moreover, mating frequency declined in presence of DATS compared with the controls. The percentage of females engaging in calling behavior decreased significantly with time, to less than 50%, 2 days after treatment, while a high calling percentage (>80%) was recorded for control moths on all 4 days. In addition, DATS had an inhibitory effect on the mating duration of *S. cerealella*. Moreover, a significant reduction was observed in the amount of sex pheromones extracted 8 h and 9 h after treatment. Our findings suggested that DATS has the potential to manipulate the moth population at LC_20_ and would be an efficient alternative to synthetic insecticides for the control of pests having low toxicity to non-target organisms and ecosystems.

## 1. Introduction

Larvae of the Angoumois grain moth, *Sitotroga cerealella* (Olivier) (Lepidoptera: Gelechiidae), primarily feed inside stored grains, and are serious pests worldwide [1]. At present, chemical controls are the most effective tools for pest management in stored grains. The chemical control of *S. cerealella* is unsatisfactory, due to chemical resistance and resulting residues in stored grains [2,3]. Due to the increasing demand for pesticide-free crops and foods and to discourage the applications of synthetic insecticides, there is an urgent need to find alternative methods to those based on conventional pesticides [4].

Recently, behavioral manipulation, such as manipulation of reproductive behavior has received widespread attention to control some crop/wood-boring pests [5]. Reproductive behavior, which integrates behavioral and physiological events in both sexes, including sex recognition, courtship, mating, and female post-mating behavioral changes, can be influenced by the environment [6]. In insects, mating behavior is an important process for reproduction. Mostly female moths attract males via release of species-specific sex pheromones [7] and then males searching for potential mates detect the chemical cues around them with their antennae [8]. After mate recognition, the male displays pre-mating status and copulates. For most insect species, the release of sex pheromones, calling, and mating display circadian rhythmic patterns [9]; a large number of moth species mate during the scotophase; and calling behavior is related to their unique female sex pheromone-releasing rhythm [10]. Some studies showed that mating behavior, frequency, and duration can directly influence insect reproductive fitness and ability [11,12].

Considering the characteristics of *S. cerelalla*, which feeds inside the grains during the larval stage, [3], the reproductive behavioral manipulation provides a new mechanism for their control using plant materials (plant essential oils and their bioactive chemical components) with high insecticidal activity [13,14]. In recent years, many studies have proved that plant resource materials with antifeedant, repellent, or insecticidal action hold the feasibility for the control of stored-product pests. Likewise, our previous studies have proven that garlic essential oil and its two active substances, diallyl disulfide (DADS) and diallyl trisulfide (DATS), can greatly reduce the fecundity of *S. cerealella* females [15]. The reduction in oviposition indicates one or more adverse effects, including effects on mate locating, courtship, oviposition, or associated physiological events [16]. Our previous molecular studies showed that the major garlic substances block mate recognition between the sexes, which in turn leads to the disruption of mating behavior [17]. The real reason why major components of garlic essential oil regulate the reproductive behavior of both sexes and reduce female fecundity is still unclear.

The purpose of this study was to: (1) determine the effect of DATS, a major component of garlic essential oil, on the fecundity of *S. cerealella*; (2) detect anomalies in the mating behavior of moths treated with DATS; (3) assess the influence of DATS on the circadian mating and calling rhythms of females, and mating frequency and duration; and (4) produce a preliminary estimate of the effect of DATS on the amount of sex pheromones secreted by female *S. cerealella*. Clarifying the effect of the essential oil of garlic on the reproductive behavior of *S. cerealella* will be useful not only to understand the specific objective of DATS, but will be able to provide appropriate indications on the most suitable formulation of garlic essential oil and its delivery methods.

## 2. Materials and Methods

### 2.1. Insects

The strain of *S. cerealella*, associated with wheat and maintained in a laboratory for the experimental tests with DATS, came from Wuhan, Hubei province, China, where it was collected in 2017. This population was kept at 28 ± 1 °C and 75% ± 5% relative humidity with a photoperiod of 14:10 h (light:dark). Grains were replenished and the moth colonies were inspected every week for wild moths. In order to obtain virgin adults, each grain containing a pupa was kept in small glass tubes capped with cotton cloths and held at the rearing conditions stated above. The grains containing pupae were identified by a swollen black spot. The grains were checked hourly for adult emergence, and then 1-day-old *S. cerealella* adults were collected for the experiments.

### 2.2. Garlic Compound 

The active compound of garlic essential oil, DATS (purity > 90%) was obtained from Sigma-Aldrich (Steinheim, Switzerland) and stored at 4 °C.

### 2.3. Evaluation of the Fecundity of S. cerealella after DATS Treatment

A set of ten pairs of one male and one female, both 1-day-old virgin adults of *S. cerealella*, was treated by fumigation with DATS at a dose of 0.015 µL/L in air (LC_20_). DATS was applied as 1.5 µL DATS dissolved in 985 µL acetone by a micropipette, and 10 µL of the solution was picked up and deposited on a filter paper (diameter 2.0 cm). The filter paper was air dried for 2 min to evaporate acetone. Then, the filter paper (diameter 2.0 cm) impregnated with LC_20_ of DATS (treated) was attached by gum to the inner surface of the lid of a glass tube (1000 mL, 10 cm in diameter × 13 cm in height). The lid was put onto a glass tube for fumigation. Moths were compared with sets of as many pairs of non-fumigated identical adults. Transparent glass tubes of about 1000 mL (10 cm in diameter × 13 cm in height), provided with a hermetic screw cap and kept at the rearing conditions, were used for treatments under the previously described conditions. A filter paper (diameter 2.0 cm) impregnated with LC_20_ of DATS (treated) or neutral (control) was attached by gum to the inner surface of the lids of the glass tubes. The lids were put onto glass tubes containing a folded filter paper (0.8 cm × 7 cm) as a site for egg laying. At the end of the preparation of the device for the tests, each tube contained 20 (10 males and 10 females) virgin 1-day-old adults, a filter paper as support for the same adults, and the fumigant or neutral element for the treatment or control, respectively. Four days later, the numbers of eggs laid on the filter papers were counted for each group. All eggs were then reared under the same conditions for 2 days. After 2 days, the color of eggs laid by mated females changed to pink; however, eggs laid by unmated females remained white. The number of pink and white eggs were counted under a microscope [18]. The fertilization rate was calculated as the proportion of pink eggs compared to the total number of eggs, and together with the total number of laid eggs, the fertilization rate was considered as an index of the moths’ fecundity. Three replicates were performed.

### 2.4. Observation of the Circadian Courtship Rhythm of Females and Mating Frequency of S. cerealella after DATS Treatment

Ten pairs of 1-day-old virgin *S. cerealella* adults were used to ascertain the effect of DATS on the circadian mating rhythm, using the same fumigation method described above, with half of the tubes being controls. Observations were made every half-hour for 4 consecutive days to record the number of mating moths (the circadian courtship rhythm of females was observed at scotophase using a 30 W red-light flashlight). Devices containing untreated filter papers were used as controls. Five replicates were set up.

To better understand the mating frequency after treatment with a sublethal dose of DATS, the females were dissected under a microscope at 10 × magnification to determine the presence and number of spermatophores in the bursa copulatrix on days 1, 2, 3, and 4 after fumigation. Fine spermatophores in the bursa copulatrix of mated female were observed. The mating frequency of each moth was calculated as the number of spermatophores in the bursa copulatrix. Three replicates were set up for all the experimental groups.

### 2.5. Determination of the Periodicity and Percentage of Calling Behavior of S. cerealella after Treatment

A 1-day-old virgin *S. cerealella* female was introduced into a glass cylinder of about 1000 mL (10 cm in diameter × 13 cm in height) with a filter paper impregnated with LC_20_ of DATS (treated) or neutral (control), as described above. Female calling behaviour was observed every half-hour for 4 consecutive days using a 30 W red-light flashlight to observe calling periodicity at scotophase. Three replicates of 10 insects each were performed for each group. The calling behaviors were classified as a female in a stationary posture with the abdomen thrusting upward and protracting the last abdominal segments or “ovipositor” to actively release sex attractants [19] at a specific time (hourly) and during experimental days.

### 2.6. Observation of Mating Duration of S. cerealella after DATS Treatment

One day post-emergence, a random male and female were paired in a 1 L glass device. The treatment methods and conditions were as above. To record the onset and termination of mating, observations were conducted at half-hour intervals over a 24 h period for 4 consecutive days using a 30 W red-light flashlight to observe mating behavior at scotophase. Each treatment was replicated 45 times. Insects that died before day 4 were excluded.

### 2.7. Effect of DATS on the Amount of Sex Pheromones Secreted by Female S. cerealella

Twenty 1-day-old virgin females were fumigated with DATS at LC_20_ in a 1 L glass device when coming into the dark period of the photoperiod. Sex pheromone was extracted 2, 7, 8, and 9 h after fumigation according to the method described by Ma et al. [16]. Briefly, pressure was applied to the abdomen of female moths until the ovipositors (8th and 9th abdominal segments) extruded. Then, 20 ovipositors were excised and soaked in 200 μL of hexane for 30 min. The extracts were filtered and concentrated to 100 μL with nitrogen, and 3 μL of extracts per sample was analyzed using Gas chromatograph mass spectrometers (GC-MS). The GC-MS instrument was operated following the method of Ma et al. [16]. The sex pheromone of *S. cerealella* (7Z,11E-hexadecadien-1-ol acetate) was the component at the retention time of 12.4 min. Then, the amount of sex pheromone between different treatments was compared based on the area of sex pheromone extract at a retention time of 12.4 min. Five replicates were performed for each treatment.

### 2.8. Statistical Analysis

The effect of DATS on the fecundity, mating percentage, calling rate, mating duration, and amount of sex pheromone was analyzed using an independent-samples *t*-test by comparing the mean of the treatment and control group (*p* < 0.05) using SPSS v19.0 (SPSS Inc., Chicago, IL, USA). The peak of mating was calculated using the equation:
T = ∑(Ti × Ni)/Ni,
where T is the peak of mating in scotophase, Ti is the observation time at half-hour intervals, and Ni is the number of mating pairs observed at half-hour intervals. Furthermore, a covariance analysis was performed to analyze the rate of mating frequency. Variations in the calling rate between the treatments on different treatment days were analysed using one-way ANOVA (Duncan multiple range test, DMRT) with SPSS.

## 3. Results

### 3.1. Efficacy of DATS on the Fecundity of S. cerealella

A significant reduction in the number of eggs and percent fertilization rate was recorded between control and DATS-treated females. The number of eggs laid by females treated with DATS was significantly lower than that laid by control females (Figure 1A). Fertilization rates were significantly lower for DATS-treated females than for control females (Figure 1B). Moths that died during the experiment were excluded from analysis, so the reduction in egg laying was not caused by the death of moths. These results further demonstrated that DATS possessed inhibitory oviposition activities at the sublethal dose (LC_20_).

### 3.2. Effect of DATS on the Circadian Mating Rhythm and Frequency of Female S. cerealella

The mating behavior of *S. cerealella* was observed during scotophase. For both the control and treatment groups, one distinct high peak was associated with mating activity on the first day; however, the peak time was different. The peaks were observed at 7 h and 4 h into the scotophase for the control and treatment groups, respectively (Figure 2A). Prior to 4 h into scotophase, the mating frequency of the control group was slightly lower than that of the treated group, particularly for the 2nd hour of scotophase; there was a significant difference in mating frequency between the groups (*p* < 0.05). After 4 h, the mating frequency continued to rise in the control group, but it began to decrease in the treated group. At the 7th, 8th, and 9th hour into scotophase, mating frequency tended to be significantly higher in the control group than for the treated group during the 7th to 9th hours. The mating frequency of control females was 86.00 ± 5.10%, which was significantly higher than the treated group (53.17 ± 7.01%) (Figure 3A). On day 2, mating was delayed by 2 h compared to day 1 (Figure 2B). The highest mating frequency was recorded at 9 h into the observation period. There was a significant difference in mating frequency between the control and treatment groups at 7 h and 8 h into the observation period. On day 3 after treatment, the onset of mating shifted another 2 h for both the control and treated groups. The mating frequency of the treatment group dropped to 20.67%, while the control remained above 50% (Figure 2C). On day 4, the onset of mating did not shift, but mating frequency decreased to 19.78% and 32.00% for the treatment and control groups, respectively (Figure 2D).

The peak point of mating (h into scotophase) was significantly different between the control and DATS-fumigated groups (Table 1). The mating peak for the DATS group was significantly earlier on day 1 (*p* < 0.05), while it was later on the following three days, although it was significantly different only on days 2 and 3 (*p* < 0.05).

The mean mating frequency per day is presented in Figure 3. Both male and female *S. cerealella* had multiple mates, and the mating frequency increased over time (Figure 3). The increases in mating frequency were significantly slower for treated moths compared with the control (*p* < 0.001; Figure 3).

### 3.3. Effect of DATS on the Periodicity and Percentage of Calling of S. cerealella

The calling rhythm of female moths changed significantly after DATS treatment (Figure 4). The females exhibited the calling position soon after the scotophase began on day 1. In the first 3 h, more females showed calling posture in the DATS-treated group than the control (Figure 4A), particularly during the 2nd hour (*P* < 0.01). Between 6 to 9 h on the first day, the control group showed the calling posture more frequently than the treatment group. In the next 3 days, the calling behavior of the treated group occurred slightly later than the control (Figure 4B–D). The female calling percentage of the treated group was less than the control at all points, while a significant difference was recorded at only some time points (Figure 4B–D).

Similarly, the mean calling percentage of female moths per day after DATS treatment was calculated (Figure 5). During the experimental days, no significant difference among the calling percentages were observed in the control, which had means of 86.67 ± 3.33, 80.00 ± 0.01, 90.00 ± 5.77, and 86.67 ± 8.82, on days 1, 2, 3, and 4, respectively (Figure 5). A significant difference was observed among DATS-treated moths (*p* < 0.05) and the mean calling percentage for the first day (83.33 ± 3.33) was significantly higher than that of the second (50.00 ± 5.77), third (26.67 ± 8.82), and fourth day (16.67 ± 8.82) (Figure 5). The results indicated that there was no difference in calling percentage on the first day; however, calling percentage was significantly lower in the treatment group on the following three days.

### 3.4. Effect of DATS on the Mating Duration of S. cerealella

The daily mean copulation duration for the four experimental days was observed, and the results showed that the mating duration on the first day was longer than the following three days for both the control and treatment groups (Table 2). Additionally, on the first two days, the mating duration of the control lasted significantly longer than the DATS group (*p* < 0.05), but mating behavior was not observed on the third and fourth day in the DATS group due to the inhibitory effects of DATS. The results indicated that copulation duration of the control group was significantly longer on all days, which proved that DATS effectively reduced copulation duration and inhibited the copulation of moths.

### 3.5. Effect of DATS on the Amount of Sex Pheromone Secreted by Female S. cerealella

The amount of sex pheromone extracted 2 h after fumigation showed no significant difference in treatment and control; however, at 8 h and 9 h after treatment (Figure 6), there was a significant reduction in the treatment group relative to the control (*p* < 0.05). The variation in sex pheromone amount seemed to coincide with the changing tendency of circadian mating rhythms and calling periodicity of females on the first day.

## 4. Discussion

With the advantages of low toxicity to non-target organisms and ecosystem [20,21,22,23], and slow evolution of insect resistance [24], behavior manipulation techniques using plant-based compounds are gradually becoming an accepted tactic for integrated pest management [5,25]. Considering the important biological characteristics of *S. cerealella* and economic damage to crops and stored products, the manipulation of adult moth reproductive behavior is an important technique for pest control. Boshra (2007) [3] reported that *S. cerealella* has no direct contact with contact pesticides in the larval stage, resulting in the need for an alternate strategy to control the pest. Studies suggest that moth infestation can be managed through altering physical characteristics such as grain hardness, texture, and nutrition content of stored products, such as amylose, coumaric acid, ferulic, and phenolic content [26]. Similarly, DATS causes a reduction in total protein, total soluble sugar, and trehalase activity of moths (unpublished data), while Slanskry (1985) [27] reported that development, fecundity, and body weight were affected when the carbohydrate or protein content of grains was altered.

Edible oils (plant essential oils and extracts), containers, insecticides (spinosad), endosymbiotic bacteria, and garlic derivatives are used to reduce *S. cerealella* infestation in stored grains [28]. Garlic essential oil is mainly composed of DADS and DATS, which are easily extracted and possess insecticidal activity against *S. cerealella* [15]. Our previous studies demonstrated that garlic essential oil and its active substances inhibit oviposition in *S. cerealella* at a high fumigant dose (LC_50_), above 1 μL/L [15]. Interestingly, when the fumigant concentration was reduced to LC_20_, DATS and DADS showed properties to inhibit the oviposition by the moths. It is well known that mating can stimulate the ovipositional behavior of insects [29]. In other words, egg reduction in the moth is due to a variation in reproductive behavior after DADS and DATS treatment. To verify our hypothesis, we conducted this study. The results showed that DATS can cause a reduction in mating frequency and duration, and thereby decrease the percentage of fertilized eggs laid by females. Through our previous study, we found that both male and female *S. cerealella* mate with multiple partners [15]. DATS fumigation significantly reduced female mating frequency and mating duration, and this may be one of the reasons why DATS reduced the number of eggs laid by females. Compared with single mating, multiple mating can not only significantly enhance the fecundity of female *Gryllus velitisa* (Alexander and Bigelow, 1960) (Orthoptera: Gryllidae), but also prolong its life span [30]. Similar results have been found in other insect species, such as *Gryllus lineaticeps* (Stal, 1861) (Orthoptera: Gryllidae) [31]. Similarly, the fecundity of *Agrotis ypsilon* (Hufnagel, 1766) (Lepidoptera: Noctuidae) is influenced by mating frequency [32]. It is not clear whether the decline in fecundity of female *S. cerealella* is due to a reduction in mating frequency caused by DATS, or due to other factors; this needs to be explored further.

Many insect activities that occur in a daily cycle are restricted to a certain time, like mating, courtship, and calling behavior [33]. Insect mating is usually rhythmic, and different species often mate only at specific times. Here, we found that mating, courtship, and calling behavior of *S. cerealella* adults occurred during the scotophase, with no activity observed during the photophase; the behavior of *S. cerealella* presented a diurnal periodicity phenomenon. After being treated with DATS, circadian mating rhythms and calling periodicity changed significantly. In order to achieve diurnal periodicity, the production of sex pheromone must be synchronized [6]. We observed that DATS affected female sex pheromone production.

Taken together, these studies suggested that DATS regulates the adult reproductive behavior of *S. cerealella* at the sublethal concentration by reducing the mating duration and frequency, and calling percentage of females. DATS, which is easily prepared, can be used at very low doses to manipulate the reproductive behavior of *S. cerealella*.

## Figures and Tables

**Figure 1 insects-11-00021-f001:**
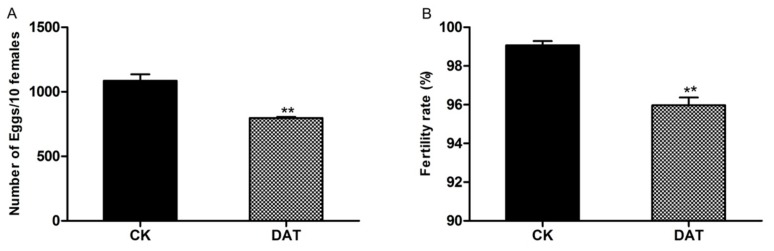
The effect of diallyl trisulfide (DATS) on (**A**) the mean number of eggs laid per day by 10 females and (**B**) the mean number of fertilized eggs, compared to control females (CK). Error bars indicate SEM and ** indicates a significant difference (*p* < 0.05, *t*-test).

**Figure 2 insects-11-00021-f002:**
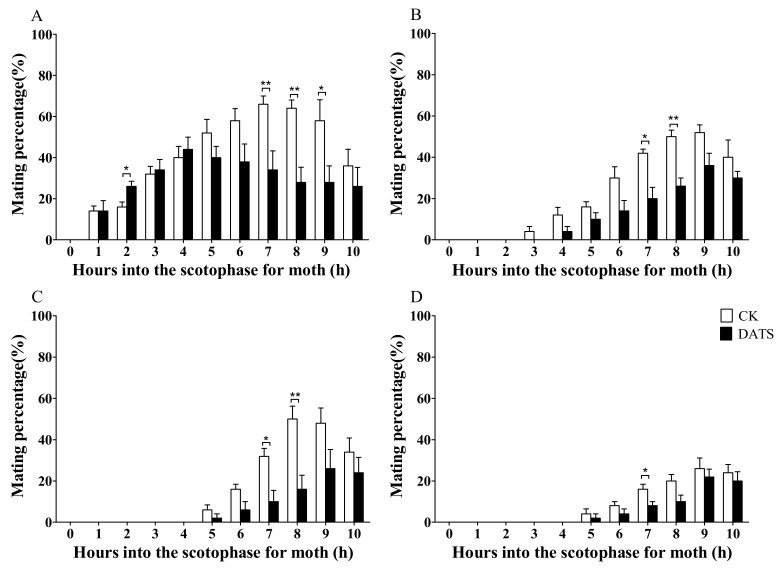
Effect of diallyl trisulfide (DATS) on the circadian mating rhythms of *Sitotroga cerealella* on (**A**) day 1, (**B**) day 2, (**C**) day 3, and (**D**) day 4. Error bars indicate SEM. Asterisks indicate significant difference when compared with the control (CK), * *p* < 0.05, ** *p* < 0.01 (*t*-test).

**Figure 3 insects-11-00021-f003:**
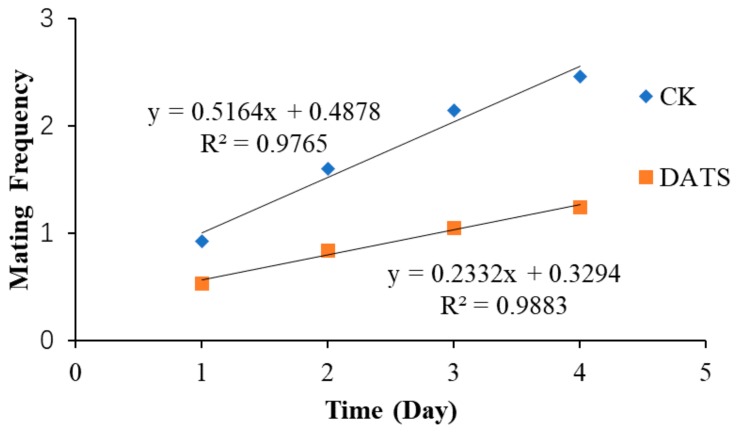
Effects of diallyl trisulfide (DATS) on the mating frequency of *Sitotroga cerealella* compared with the control (CK). The slope of each line was compared using a covariance analysis, Degree of freedom (Df) = 2, F = 22.921, *p* = 0.002.

**Figure 4 insects-11-00021-f004:**
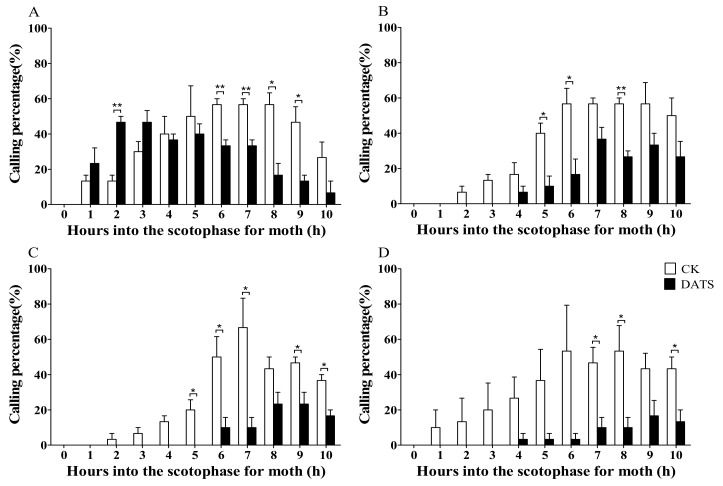
Effects of diallyl trisulfide (DATS) on the calling periodicity of *Sitotroga cerealella* on (**A**) day 1, (**B**) day 2, (**C**) day 3, and (**D**) day 4. Error bars indicate SEM. Asterisks indicate significant differences at * *p* < 0.05 and ** *p* < 0.01 (*t*-test). CK is the control group.

**Figure 5 insects-11-00021-f005:**
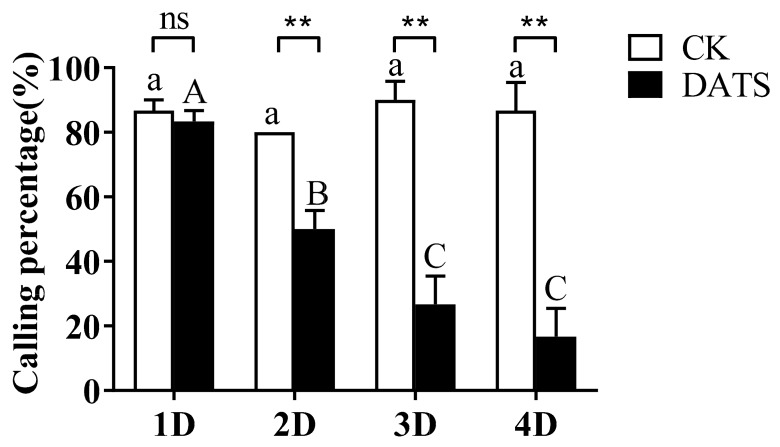
The effect of diallyl trisulfide (DATS) on the calling percentage of *Sitotroga cerealella*. Error bars indicate SEM. ** Indicates significance at *p* < 0.01, and ns indicates no significant difference (*t*-test). The same letter (lowercase for control (CK) and uppercase for DATS) indicates no significant difference in calling percentage for each treatment on different days (D) after treatment (*p* < 0.05, Duncan multiple range test (DMRT).

**Figure 6 insects-11-00021-f006:**
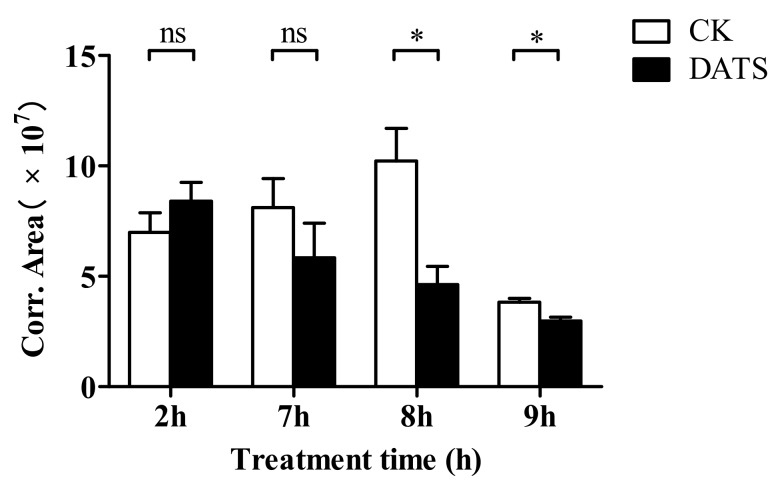
The effect of diallyl trisulfide (DATS) on the amount of sex pheromone secreted by female *Sitotroga cerealella* compared to the control (CK). * Indicates significance at *p* < 0.05, while ns indicates no significant difference (*t*-test).

**Table 1 insects-11-00021-t001:** Effects of diallyl trisulfide (DATS) on peak of mating of *Sitotroga cerealella* where CK is the control group.

Treatment	Peak of Mating (Hours into the Scotophase)			
	Day 1	Day 2	Day 3	Day 4
CK	6.20 ± 0.15	7.53 ± 0.10	8.01 ± 0.14	8.20 ± 0.05
DATS	5.32 ± 0.30 *	8.04 ± 0.17 *	8.76 ± 0.25 *	8.62 ± 0.23

* Indicates significance at *p* < 0.05, *t*-test.

**Table 2 insects-11-00021-t002:** Effect of diallyl trisulfide (DATS) on the copulation duration of *Sitotroga cerealella*.

Treatment	Copulation Duration (h)			
	Day 1	Day 2	Day 3	Day 4
CK	4.58 ± 0.25	2.39 ± 0.24	2.30 ± 0.22	2.36 ± 0.20
DATS	3.59 ± 0.40 *	1.45 ± 0.32 *	-	-

Values are presented as mean ± SEM. Data comparing groups in each column followed by * are significantly different at *p* < 0.05 (*t*-test). CK refers to the control group and - indicates that no copulation occurred.

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
