# Peer review of "Effect of Diallyl Trisulfide on the Reproductive Behavior of the Grain Moth, Sitotroga cerealella (Lepidoptera: Gelechiidae)"

_insects, 2019, doi:10.3390/insects11010021_

Round 1

Reviewer 1 Report

This manuscript describes experiments aimed at determining the effects of a compound in garlic oil, diallyl trisulfide, on the mating behaviors and fecundity of a stored products pest, the Angoumois grain moth.  The experimental treatments did produce some effects, but there are several places in which crucial experimental detail is missing.  Specific comments as follows:

L 19, L 88 and elsewhere: the LC20 dose is listed as 0.015 ul/L, but it is entirely unclear what these units refer to.  Do you mean 0.015 ul per liter of air, or was this 0.015 ul per liter of carrier solvent, or…?  Without clearly and explicitly stating what this concentration refers to, it is impossible to interpret the results. L 37, clarify that it is the larvae that infest the grain, not the moths. L 39-40: why is chemical control unsatisfactory, because of resistance, or difficulty in applying to grains, or…? Clarify L 44, do larval stages exhibit reproductive behaviors? Clarify wording here. L 47 and L 266: this statement needs to be modified because there are many other crucial physiological processes involved in reproduction L 61, are there only two active substances in the oil, or are there more but this team has only worked with two of them. Clarify Section 2.1, provide more detail about rearing the insects, including how long they have been in colony, whether the colony has been periodically refreshed with wild type individuals, what grain or other substrate the insects were reared on, rearing containers/conditions for the lab cultures, how infested kernels were identified so that they could be separated to allow adults to emerge, etc. L 88-90 states that a filter paper was impregnated with the test compound at 0.015 ul/L, but it does not describe the solvent that was used as a carrier, i.e., I presume that it would not be possible to measure and apply 0.015 microliters, or 15 nanoliters of the test compound. This dose also seems very low, i.e., it would be about 15 micrograms.  Is this indeed correct, or have the authors miscalculated the amount applied.  Do they really mean that 15 microliters were applied per treatment, or roughly 15 milligrams, or….? It is obviously crucially important to know accurately what the applied dose was, and exactly how it was applied. L 93 states that untreated filter papers were used as controls. However, the controls should have been treated with the same dose of solvent that was used to apply the test compound in the treated papers. L 88 on, were the moths provided with any food or water in these sealed jars? For all the experiments described in sections 2.3-2.7, explicitly state the light and temperature conditions, and the light dark cycle. L 115 and elsewhere, what exactly is meant by “courtship”? The authors have provided a clear definition of the calling behavior, but it is entirely unclear what they mean by courtship?  Do they mean copulation, or…?  Please clarify, because this variable is used in several of the figures. L 121, provide a reference for this calling behavior. Section 2.7 on pheromone content: the authors state that pheromones were analyzed 2,7,8, and 9 hours after fumigation, but it is entirely unclear where these times were in relation to the light/dark cycle, which is crucially important for calling behavior. They need to state explicitly how these extraction times relate to the L/D cycle, and why these particular times were chosen.  The authors also need to provide a concise description of how the pheromone extracts were prepared, how they were analyzed, and what exactly was measured, i.e., did they measure one component, or two components, or…? L 154 states that no moth mortality was observed over the 4-day course of the experiment, using a treatment dose corresponding to the LC20. However, if indeed the dose applied corresponded to the LC20, then shouldn’t 20% of the moths have died during this time period?  Does this suggest that the miscalculated the dose and applied less than the LC20, or ….?  Along these lines, it would be useful to have a clear and explicit definition of exactly what they mean by LC20, i.e., what was the time period used to calculate the LC20, and was the LC20 calculated under exactly the same experimental conditions as were used in the experiments described here? L 158, replace “active substances from garlic essential oil” with the compound that was actually used, i.e., “diallyl trisulfide”. L 200 on, the authors have not stated what they mean by courtship, and it makes it very confusing. For example, on L 203 they state that Fig 4A shows females exhibiting calling, but L 204 then talks about courtship, not calling. Then further on, they state that Figs 4B-D show calling behavior, but in the next sentence they are again talking about courtship behavior.  In addition, the y axis labels in Figs 4A-D are Courtship percentage, despite the fact that the text says they show calling behavior. So, what exactly was measured (calling behavior or courtship by males, or….?), and what exactly do these figures show (calling behavior or courtship, or….?). It is unclear how the data in Fig 5 were collected, and what exactly this data is supposed to represent. Does it show the total percentage of moths that mated at least once per day, or the total number of females that called at least once per day, or…? Please clarify, it is again very confusing. Table 2, does this show the mean duration of each copulation, or the mean time spent in copulation per day. L 240: the treatment was not slightly more than the control as stated; there was no significant difference between the two, so they were effectively the same. Correct please.  Also, as stated above, please state exactly what was measured and how. L 249, the authors state that DATS has “no toxicity to nontarget organisms and vertebrates”. However, the safety data sheet for DATS states:

Causes serious eye irritation.

Material may be irritating to the mucous membranes and upper respiratory tract.

May be harmful by inhalation, ingestion, or skin absorption.

May cause drowsiness and dizziness.

May cause skin or respiratory system irritation.

Repeated exposure may cause skin dryness or cracking.

To the best of our knowledge, the toxicological properties have not been thoroughly investigated

Thus, there are indeed some safety concerns with DATS, and what would be better would be to provide a more careful description of the possible hazards associated with DATS, and how those may or may not be relevant to the use of this material as a grain fumigant.  In addition, does this compound have a strong odor or taste, that might contaminate and downgrade or ruin any grain that is treated with DATS?

L 249, what reason do the authors have to believe that the insects would be slow to evolve resistance to DATS, given how successful insects have been at rapidly developing resistance to many other insecticides, and some behavior modifying compounds? L 264, provide a reference to the statement that previous research showed that both sexes mate multiple times. Same with L 280, provide references. Interpretation of Table 1: the DATS treatment caused females to call significantly earlier on day 1, but significantly later on days 2 and 3, and then no difference between DATS treatment and control on day 4. What is the interpretation of this inconsistent pattern? Fig 4 and Fig 2, crucially important for the discussion, the authors have neglected to point out that in the controls, on days 3 and 4 especially, some control females apparently called for several hours (Fig 4) before any mating occurred (Fig 2), so this was wasted calling effort. In contrast, for the treatment females, calling and mating seemed to be much better synchronized, i.e., on days 3 and 4, as soon as females started to call, mating started to occur. Comments? L 285, the authors make the point that for mating to occur efficiently, calling and mating must be synchronized. However, as stated above, comparing the data in Figs 2 and 4, it seems as though calling and mating are synchronized BETTER in the treatment group than in the control group, with less wasted calling effort by females. Comments? L 286 -305 is entirely speculative and should be removed because the authors have no idea as to the mechanism of action of DATS, i.e., whether it is affecting hormone levels, or whether it is affecting biosynthetic enzymes involved in production of the pheromone, or whether it is making the moths sick so that they don’t want to call, or …? Without having some idea as to the mode of action of DATS, it is impossible to say or even reasonably speculate on how exactly it may be affecting pheromone production. L 294, PBAN controls pheromone production, rather than enhancing it. L 306: DATS is not an oviposition inhibitor, and should not be described as such. As the authors have shown, it appears to act upstream of oviposition, so that treated moths have fewer fertile eggs to oviposit.  To claim that it is an oviposition inhibitor, the authors would have to divide a mated cohort of moths in two, treat half with DATS with the other half as untreated controls, and then see whether the numbers of eggs laid were different. This paper has been written as though there were no other research investigating the use of essential oils and similar products for control of this pest, whereas this appears to be an area of very active research, see the review Bushra, S. and Aslam, M., 2014. Management of Sitotroga cerealella in stored cereal grains: a review. Archives of Phytopathology and Plant Protection, 47(19), pp.2365-2376. Thus, it might be useful for the authors to provide some context for their work by concisely describing some of the related work that is going on with this pest. I have considerable sympathy with the fact that English is clearly not the native language of the authors, but I would strongly recommend that the manuscript be thoroughly edited and proof read by a native English speaker or professional editing service because in some cases, it is difficult to understand the sense of what the authors are trying to say.

Author Response

Respected reviewers,

We are grateful to the reviewers for giving time to review our manuscript. We deeply appreciate the reviewer’s valuable comments and suggestions to improve our manuscript. We tried our best to improve the quality of manuscript in light of reviewer’s comments; meanwhile, the manuscript was polished by a professional scientific editing service (Wiley Editing Services). We hope that the revised manuscript will meet the standards of publication in your good journal.

The point-by-point answers and explanations for all comments were listed below.

Note: Reviewer#1 responses are highlighted with Green where possible in the revised manuscript.

Reviewer #1;

Comment 1- L 19, L 88 and elsewhere: the LC20 dose is listed as 0.015 ul/L, but it is entirely unclear what these units refer to.  Do you mean 0.015 ul per liter of air, or was this 0.015 ul per liter of carrier solvent, or…?  Without clearly and explicitly stating what this concentration refers to, it is impossible to interpret the results.

Response- Thanks for your kind comment. Yes, our dose LC20 means 0.015µL per liter of air. The treated method was as following: 1.5µL DATS was dissolved into 985µL acetone by a micropipette, 10µL of the solution was picked up and deposited on a filter paper (diameter 2.0 cm). The filter paper was air dried for 2 minutes to evaporate the acetone. And then the filter paper (diameter 2.0 cm), impregnated with LC20 of DATS (treated) was attached by gum to the inner surface of the lid of a glass tube (1000 ml, 10 cm in diameter x 13 cm in height). The lid was put onto a glass tube for fumigation. Concentration for fumigation was 0.015 μL/L air. The same method was followed by Huang et al. (Huang et al. 2000) and Yang et al. (Yang et al. 2012).

During the experiments, considering there will be adverse side effect of acetone on adult moths, so we evaluated and confirmed that there was no difference between the groups treated with 10µL acetone alone and the blank controls without any treatment for the adults’ No. eggs laid, mortality rate, and mating frequency within 24h. Please see the Fig. 1. Hence, we considered the blank controls without any treatment as “Controls” compared with DATS-treated groups in the whole experiment.

Fig.1 Effects of acetone on the total number eggs laid, mortality rate and mating frequency of adult moths 48 hrs after treatment

Note: CK means groups without any treatment/ blank controls and BT means acetone-treated groups in Fig.1.

Huang Y, Chen SX, Ho SH (2000) Bioactivities of methyl allyldisulfide and diallyl trisulfide from essential oil of garlic to two species of stored-product pests, Sitophilus zeamais (Coleoptera: Curculionidae) and Tribolium castaneum (Coleoptera: Tenebrionidae). J Econ Entomol, 93(2):537–543.

Yang FL, Zhu F, Lei CL (2012) Insecticidal activities of garlic substances against adults of grain moth, Sitotroga cerealella (Lepidoptera: Gelechiidae). Insect Sci, 19:205-212.

Comment 2- L 37, clarify that it is the larvae that infest the grain, not the moths.

Response- Thank you so much for your kind suggestion, the information is added in Line 35. The female moths lay eggs on the grains and the larvae grow inside the grain, feed the content of the grain, after completing pupation, the adult moths emerge from the grain.

Comment 3- L 39-40: why is chemical control unsatisfactory, because of resistance, or difficulty in applying to grains, or…?

Response- Thank you so much for your expert comment. During larval and pupal stage, the pest resides inside the grain and larval stage is considered the most destructive, because it feeds inside contents of grain, to control the larvae through synthetic pesticides, systemic insecticides need to apply to the grains which causes residue in food and environmental pollution. Papers (Aziz El and Shadia EA 2011; Holland PT 1994) reported that several synthetic insecticides like pirimiphos-methyl residue remained constant, organochlorine and synthetic pyrethroids residue also remain very stable in stored grains. The pesticides also cause resistance to stored product pests (Boshra et al. 2007; Kaarag and sakine 2015). The information is added to manuscript in the revised manuscript (Line 38).

Comment 4- Clarify L 44, do larval stages exhibit reproductive behaviors? Clarify wording here.

Response- Thanks for your expert comment, in the revised manuscript (L122-123), we modified the sentence, we are very sorry that we cannot clearly expressed the sentence, there we want to write about the behavior manipulation (not actually mean reproductive behavior manipulation in larval stage) for the pest control.  

Comment 5- L 47 and L 266: this statement needs to be modified because there are many other crucial physiological processes involved in reproduction

Response- Thank you so much for your suggestion, the statement was modified in the revised manuscript and changed the description in discussion section. 

Comment 6- L 61, are there only two active substances in the oil, or are there more but this team has only worked with two of them.

Response- Thanks for your kind comment, Garlic essential oil contains sulphur containing compounds (alliin, allicin, diallyl sulfides, diallyl disulphide, diallyl trisulphide), vitamins (A, B complex and C) enzymes, amino acids, proteins and flavonoids.

Our team screened these sulphur containing compounds from garlic essential oil, respectively tested their bioactivity against stored-product pests, and proved the two active substances, diallyl disulfide and diallyl trisulfide(DADs and DATs), in the oil possessing high insecticidal activities against stored-product pests. Hence, our team is just working on the two main compounds from garlic oil.

Comment 7- Clarify Section 2.1, provide more detail about rearing the insects, including how long they have been in colony, whether the colony has been periodically refreshed with wild type individuals, what grain or other substrate the insects were reared on, rearing containers/conditions for the lab cultures, how infested kernels were identified so that they could be separated to allow adults to emerge, etc.

Response- we are thankful to the respectful reviewer for pointing out this mistake. In the revised manuscript the information has been included as per your valuable comments in the revised manuscript (Line 157-164).

Comment 8- L 88-90 states that a filter paper was impregnated with the test compound at 0.015 ul/L, but it does not describe the solvent that was used as a carrier, i.e., I presume that it would not be possible to measure and apply 0.015 microliters, or 15 nanoliters of the test compound. This dose also seems very low, i.e., it would be about 15 micrograms.  Is this indeed correct, or have the authors miscalculated the amount applied.  Do they really mean that 15 microliters were applied per treatment, or roughly 15 milligrams, or….? It is obviously crucially important to know accurately what the applied dose was, and exactly how it was applied

Response- Thanks for your kind comment. Yes, our dose LC20 means 0.015µL per liter of air. The treated method was as following: 1.5µL DATS was dissolved into 985µL acetone by a micropipette, 10µL of the solution was picked up and deposited on a filter paper (diameter 2.0 cm). The filter paper was air dried for 2 minutes to evaporate the acetone. And then the filter paper (diameter 2.0 cm), impregnated with LC20 of DATS (treated) was attached by gum to the inner surface of the lid of a glass tube (1000 ml, 10 cm in diameter x 13 cm in height). The lid was put onto a glass tube for fumigation. Concentration for fumigation was 0.015 μL/L air.  We also explained the methodology in response of your kind comment 1.

Comment 9- L 93 states that untreated filter papers were used as controls. However, the controls should have been treated with the same dose of solvent that was used to apply the test compound in the treated papers.

Response- We are very grateful to the reviewer suggestion. Of course, we had same concerned about solvent (acetone), considering there will be adverse side effect of acetone on adult moths, so we evaluated and confirmed that there was no difference between the groups treated with 10µL acetone alone and the blank controls without any treatment for the adults’ No. eggs laid, mortality rate, and mating frequency within 48 hrs. Hence, we just took the untreated groups as the blank controls compared with the DAT-treated groups in the manuscript. So we provided the data on the comparison between blank controls and acetone treated alone in the response sheet as a supplementary file. However, we agree with your valuable suggestion about the acetone filter paper used and we will incorporate such parameter in our future research work.

Comment 10- L 88 on, were the moths provided with any food or water in these sealed jars?

Response- Considering the adult traits of the adult moth, we do not need to provide any food in these sealed jars and the jars were placed in incubator with 28±1°C, 75%±5% RH with a photoperiod of 14:10 (L:D) h.

Comment 11- For all the experiments described in sections 2.3-2.7, explicitly state the light and temperature conditions, and the light dark cycle.  

Response- All experiments were conducted at 28±1°C, a photoperiod of 14:10 (L:D) h. Thanks for your kind comment, we will complement the methods in sections 2.3-2.7 in the revised manuscript.

Comment 12- L 115 and elsewhere, what exactly is meant by “courtship”? The authors have provided a clear definition of the calling behavior, but it is entirely unclear what they mean by courtship?  Do they mean copulation, or…?  Please clarify, because this variable is used in several of the figures. L 121, provide a reference for this calling behavior.

Response- we are thankful to the respected reviewer for pointing out this mistake and we are sorry that we didn’t make clear, courtship is the sum of calling behavior of whole day. We checked the calling periodicity of the moth every after half hour for 4 consecutive days, the daily calling behavior are plot as calling percentage in Figure 5. And we replaced all “courtship” in the manuscript with “calling” in the revised manuscript.

The information has been added to the revised manuscript (L315-323). Reference have been added in revised manuscript (L322).

Comment 13- Section 2.7 on pheromone content: the authors state that pheromones were analyzed 2,7,8, and 9 hours after fumigation, but it is entirely unclear where these times were in relation to the light/dark cycle, which is crucially important for calling behavior. They need to state explicitly how these extraction times relate to the L/D cycle, and why these particular times were chosen.  The authors also need to provide a concise description of how the pheromone extracts were prepared, how they were analyzed, and what exactly was measured, i.e., did they measure one component, or two components, or…?

Response- we are thankful to the respected reviewer for pointing out these weaknesses. Our sampling time depends strictly on the photoperiod. In other words, we fumigated the moth just before into the scotophase. The reason we chose to analyze the sex pheromone content at 2, 7, 8, and 9 hours after fumigation was because we found significant differences in calling percentage between the control and DATS treatment groups at these specific time points. Last, the sex pheromone of S. cerealella had only one primary component. According to the method described by Ma et al. (16), the volatile extracts were analyzed by GC-MS. The component identified at the retention time of 12.4 min was the sex pheromone of S. cerealella. We have modified the material method and added a concise description of how the pheromone extracts were prepared and analyzed in the revised manuscript (Line 471-479).

Comment 14- L 154 states that no moth mortality was observed over the 4-day course of the experiment, using a treatment dose corresponding to the LC20. However, if indeed the dose applied corresponded to the LC20, then shouldn’t 20% of the moths have died during this time period?  Does this suggest that the miscalculated the dose and applied less than the LC20, or ….?  Along these lines, it would be useful to have a clear and explicit definition of exactly what they mean by LC20, i.e., what was the time period used to calculate the LC20, and was the LC20 calculated under exactly the same experimental conditions as were used in the experiments described here?

Response- we are thankful to the respected reviewer for pointing out this mistake and we are sorry that we didn’t make clear, we accurately used the 0.015ul/L dose which caused the mortality during 4 days. Here we mean that the reduction in egg laying was not caused by the death of moths, because the died moths were excluded from the experiment. In another study that we submitted to another journal, the researcher observed 20% mortality using same dose 0.015ul/L. for academic morality we cannot reuse them here. In revised manuscript the sentence was changed in order to make it clearer to reader (L497-498).

Comment 15- L 158, replace “active substances from garlic essential oil” with the compound that was actually used, i.e., “diallyl trisulfide”.

Response- Respected reviewer, the information was replaced according to your expert suggestion in the revised manuscript (Line 510).

Comment 16- L 200 on, the authors have not stated what they mean by courtship, and it makes it very confusing. For example, on L 203 they state that Fig 4A shows females exhibiting calling, but L 204 then talks about courtship, not calling. Then further on, they state that Figs 4B-D show calling behavior, but in the next sentence they are again talking about courtship behavior.  In addition, the y axis labels in Figs 4A-D are Courtship percentage, despite the fact that the text says they show calling behavior. So, what exactly was measured (calling behavior or courtship by males, or….?), and what exactly do these figures show (calling behavior or courtship, or….?). It is unclear how the data in Fig 5 were collected, and what exactly this data is supposed to represent. Does it show the total percentage of moths that mated at least once per day, or the total number of females that called at least once per day, or…? Please clarify, it is again very confusing.

Response- we are thankful to the respected reviewer for pointing out this mistake and again we are sorry that we didn’t make clear. The experiments involved in this research were mainly focused on the calling behavior of females. We have modified the "courtship" used incorrectly in the whole paper. In Fig. 5, we counted the total number of females that called at least once per day, and then calculate the percentage of calling.

Comment 17- Table 2, does this show the mean duration of each copulation, or the mean time spent in copulation per day.

Response- Respected reviewer, the data shown in the Table 2 mean the duration of each copulation.

Comment 17- L 240: the treatment was not slightly more than the control as stated; there was no significant difference between the two, so they were effectively the same. Correct please.  Also, as stated above, please state exactly what was measured and how (This one is also about pheromones)

Response- thanks to the respected reviewer for correcting our description of experiments results. The information was corrected according to your expert suggestion in the revised manuscript (Line596-597). And we had amended the section 2.7 according to your comment 13 in the revised manuscript (Line 470-479).

Comment 18- L 249, the authors state that DATS has “no toxicity to nontarget organisms and vertebrates”. However, the safety data sheet for DATS states: Causes serious eye irritation. Material may be irritating to the mucous membranes and upper respiratory tract. May be harmful by inhalation, ingestion, or skin absorption. May cause drowsiness and dizziness. May cause skin or respiratory system irritation. Repeated exposure may cause skin dryness or cracking. To the best of our knowledge, the toxicological properties have not been thoroughly investigated. Thus, there are indeed some safety concerns with DATS, and what would be better would be to provide a more careful description of the possible hazards associated with DATS, and how those may or may not be relevant to the use of this material as a grain fumigant.  In addition, does this compound have a strong odor or taste, that might contaminate and downgrade or ruin any grain that is treated with DATS?

Response- we are thankful to the respected reviewer for pointing out this mistake and also pointed out by another respected reviewer. We agree with both honorable reviewers and we changed the sentence in the revised manuscript (Line 607) from no toxicity to low toxicity on the basis of published data, which we put the references.

Yes, DATS have an odor and in current study we separately fumigated moths with DATS in glass jars but the study was conducted in our laboratory in the rice, however the author, have no observation that DATS fumigation ruin or downgrade the quality of treated grains (Yang et al 2010). However, we agree with your expert suggestion and we will include these parameters in our future studies for the possible contamination and effects of DATS on grains.

Yang FL, Zhu F, Lei CL(2010) Garlic essential oil and its major component as fumigants for controlling Tribolium castaneum (Herbst) in chambers filled with stored grain. J Pest Sci, 83: 311-317.

Comment 19- L 249, what reason do the authors have to believe that the insects would be slow to evolve resistance to DATS, given how successful insects have been at rapidly developing resistance to many other insecticides, and some behavior modifying compounds?

Response- we are very thankful to the honorable reviewer for comment, we do believe that plant extracted compounds (botanicals) possess many target points against insect pest, not a single target like synthetic pesticides. We did transcriptomic analysis of moth treated with Lc20=0.015ul/L and found that many genes belonging to different pathways are affected by DATS. In another study we found that DAT interfered development related genes and have the effect on moth energy metabolism. Taken all together, we believe that for moth it would be difficult compared with synthetic chemicals to adopt or evolve resistance to DAT rapidly.

Comment 20- L 264, provide a reference to the statement that previous research showed that both sexes mate multiple times.

Response- Respected reviewer, the reference has been added in the revised manuscript (Line 633), according to your suggestion.

Comment 21- Same with L 280, provide references.

Response- Respected reviewer, thanks for pointing out our mistake, we are very sorry that didn’t make clear, actually in the sentence we mean the current study not the previous which we rectified in the revised manuscript (Line 640).

Comment 22- Interpretation of Table 1: the DATS treatment caused females to call significantly earlier on day 1, but significantly later on days 2 and 3, and then no difference between DATS treatment and control on day 4. What is the interpretation of this inconsistent pattern?

Response- thanks to the respected reviewer for your question. The DATS treatment caused females to call significantly earlier on day 1 may results from its similar or even higher binding affinity to ScerPBP than S. cerealella sex pheromone, 7Z, 11E-hexadecadien-1-ol acetate described by Ma et al (2016). However, continuous fumigation can cause disturbances in the olfactory pathways of S. cerealella. As a result, females called significantly later on days 2 and 3.

Comment 23- Fig 4 and Fig 2, crucially important for the discussion, the authors have neglected to point out that in the controls, on days 3 and 4 especially, some control females apparently called for several hours (Fig 4) before any mating occurred (Fig 2), so this was wasted calling effort. In contrast, for the treatment females, calling and mating seemed to be much better synchronized, i.e., on days 3 and 4, as soon as females started to call, mating started to occur. Comments?

Response-Respected reviewer, these two experiments were performed under the different conditions. The mating rhythm of moth was observed after the males and females were paired, but the calling rhythm was conducted by the females alone. Both experiments lasted for four days. In the mating rhythm experiment, most of the female were mated on days 3 and 4. But, the females in the calling experiment are virgins. So, we think there is nothing comparable between the two experiments due to different experimental conditions.

Comment 24- L 285, the authors make the point that for mating to occur efficiently, calling and mating must be synchronized. However, as stated above, comparing the data in Figs 2 and 4, it seems as though calling and mating are synchronized BETTER in the treatment group than in the control group, with less wasted calling effort by females. Comments?

Response- thanks to the respected reviewer for your question. We explained in detail in the response of previous comment. These two experiments were performed under the different conditions. The mating rhythm of moth was observed after the males and females were paired, but the calling rhythm was conducted by the females alone. Both experiments lasted for four days. In the mating rhythm experiment, most of the female were mated on days 3 and 4. But, the females in the courtship experiment are virgins. So, we think there is nothing comparable between the two experiments due to different experimental conditions.

Comment 25- L 286 -305 is entirely speculative and should be removed because the authors have no idea as to the mechanism of action of DATS, i.e., whether it is affecting hormone levels, or whether it is affecting biosynthetic enzymes involved in production of the pheromone, or whether it is making the moths sick so that they don’t want to call, or …? Without having some idea as to the mode of action of DATS, it is impossible to say or even reasonably speculate on how exactly it may be affecting pheromone production. L 294, PBAN controls pheromone production, rather than enhancing it

Response- we are very thankful to the honorable reviewer for comment and for reviewer satisfaction we deleted the sentences, as per suggestion of reviewer.

Comment 26- L 306: DATS is not an oviposition inhibitor, and should not be described as such. As the authors have shown, it appears to act upstream of oviposition, so that treated moths have fewer fertile eggs to oviposit.  To claim that it is an oviposition inhibitor, the authors would have to divide a mated cohort of moths in two, treat half with DATS with the other half as untreated controls, and then see whether the numbers of eggs laid were different

Response- we are thankful to the respected reviewer for pointing out this mistake and we agree with reviewer suggestion, the word (ovipositional inhibitor) was replaced in the revised manuscript (Line 651-652).

Comment 27- This paper has been written as though there were no other research investigating the use of essential oils and similar products for control of this pest, whereas this appears to be an area of very active research, see the review Bushra, S. and Aslam, M., 2014. Management of Sitotroga cerealella in stored cereal grains: a review. Archives of Phytopathology and Plant Protection, 47(19), pp.2365-2376. Thus, it might be useful for the authors to provide some context for their work by concisely describing some of the related work that is going on with this pest.

Response- We are very thankful to the honorable reviewer expert suggestions, the information with references have been added to the revised manuscript (Line 607-625).

Comment 28- I have considerable sympathy with the fact that English is clearly not the native language of the authors, but I would strongly recommend that the manuscript be thoroughly edited and proof read by a native English speaker or professional editing service because in some cases, it is difficult to understand the sense of what the authors are trying to say.

Response- Thank you very much for your brief comments and valuable suggestions. Indeed, there were problems with the clarity of writing and language. All typographical or grammatical errors are corrected by professional scientific editing service (Wiley Editing Services).

With the incorporation of constructive suggestions from reviewers, the manuscript has been clearly improved. We hope that the revised manuscript will satisfy the reviewers and that you will find the paper is now suitable for publication. Thanks again for your timely consideration of our manuscript.

Dr Feng-Lian Yang

Hubei Insect Resources Utilization and Sustainable Pest Management Key Laboratory

Huazhong Agricultural University

Wuhan 430070, People’s Republic of China

Tel./Fax: +86-27-87287207

E-mail: yangfenglian@mail.hzau.edu.cn

Reviewer 2 Report

The work concerns the action of diallyl trisulfide, a component of the garlic essential oil, on the reproductive performance of the gelechiid moth Sitotroga cerealella, and so it concerns a topic of great interest in the contest of the research devoted to improvement of the IPM for harmful arthropods infesting stored products, topic that is become quite fashionable in recent years.

In relation to the importance of the species, as a primary pest of the stored grains, and to the interesting approach for its control in an IPM context, the manuscript shows many limitations both in terms of English language and terminology, as well as in terms of experimental methodology and result analysis.

As regards the interpretation of the results, let me say that significant statistical differences between the average values relating to "treatment" and those relating to an "untreated control" do not allow to emphasize the value of a new method of population control of harmful insects, if these differences are not even of absolute weight or relative weight but in this case compared to other methods of sustainable IPM

In conclusion, in my opinion, the manuscript needs a radical improvement both on the English front and on the description of the "Materials and Methods" and the articulation of the "Discussion".

The elements that according to my thinking must be corrected, integrated or improved are listed below and are also indicated in the annexed manuscript pdf.

Title

Lines 1-5 Replace: “Effect of diallyl trisulfide, an active substance from garlic essential oil, on the reproductive behavior of adults of grain moth, Sitotroga cerealella (Lepidoptera: Gelechiidae)” with: “Effect of diallyl trisulfide on the reproductive behavior of the grain moth Sitotroga cerealella (Lepidoptera: Gelechiidae)”

Abstract

Line 16 - Since Olivier, in 1789, described the Angoumois grain moth as Alucita cerealella, replace "Olivier" with: "(Olivier)" or with: "(Olivier, 1789)

Lines 17-18 – Replace: “in terms of effective population control 17 of moths,” with: “Replace with: "in terms of effective control of the moth population"

Line 19 – Replace: “at LC20 (LC20 = 0.015μL/L)” with: “at a dose of 0.015 μL/L (LC20).”

Line 20 – Replace: “when treated the moth” with: "when the moth was treated"

Line 21 – Replace: “DATS, also,” with: “DATS. Furthermore,” and: “female’s” with: “female” Line 23 – Replace: “Also,” with: “Moreover,”

Line 24 - Replace: “with the existence” with: “in presence”

Line 27 – Replace: “Additionally,” with: “In addition,”

Line 30 – Replace: “Our finding suggests that DATS has the potential to manipulate the moth population at LC20” with: “Our findings suggest that DATS at LC20 has the potential to manipulate the moth population”

Lines 31-32 – The sentence “an efficient alternative to synthetic insecticides for the control of pest without harming non-target organisms and ecosystem” is a pretty bombastic phrase, without scientific support. Which data demonstrate the selectivity of DATS towards no target and beneficial insects?

Keywords

Lines 33-34 - Taking into account that the first three keywords are already present in the title and that the editorial rules provide for an alphabetical order, replace: “Diallyl trisulfide; Sitotroga cerealella; reproductive behavior; circadian mating rhythms; courtship periodicity; inhibiting oviposition” with: "botanicals; circadian mating rhythms; courtship periodicity; garlic essential oil; inhibiting oviposition; pest management; stored product protection”

Introduction

Line 37 – Replace: “Sitotroga cerealella” with: “Sitotroga cerealella (Olivier) (Lepidopera: Gelechiidae)”. This formulation cannot be present in "Abstract" and not in the “Introduction”

Line 39 – Delete: “insect” and “systems” becouse they are redundant

Line 41 – Replace: “the applications of synthetic insecticide” with: “the applications of synthetic insecticides”

Lines 41-42 – Replace: “alternatives strategies to conventional insecticides” with: "alternative methods to those based on conventional pesticides"

Line 48 – Replace: “Mostly female moths attract the male” with: “Mostly female moths attract males”

Line 50 – Replace: “After the mate recognition, the male displayed” with: “After the mate recognition, the male displays”

Line 51 – Replace: “and copulated” with: “and copulates”

Line 52 – Replace: “a large number of moth species mated” with: " a large number of moth species mate"

Line 56 – Replace: “of the S. cerelalla which feed” with: "of S. cerealella which feeds"

Line 60 – Replace: “plant resource materials . . . possessed the feasibility” with: “plant resource materials . . . hold the feasibility”

Line 60 – Replace: “the control of stored-product insect pests” with: “the control of stored-product pests”. The term “insect” is redundant!

Line 63 – Replace: “female S. cerealella” with: “S. cerealella female”

Line 64 – Replace: “including the effects on mate-locating” with: “including those on mate-locating”

Line 65 – Replace: “While our previous molecular level studies” with: “While our previous molecular studies”

Line 67 – Replace: “the real reasons how major components” with: “the real reasons why major components”

Line 68 – Replace: “female’s fecundity” with: “female fecundity”

Line 70 – Replace: “observe the mating behavior” with: “detect anomalies in mating behavior”

Lines 73-76 – The sentence: “To elucidate the effect of garlic essential oil on the reproductive behavior of S. cerealella is of practical importance for insect control because it will not only help to clarify the specific target of DATS but also provide useful information on the most suitable formulation of garlic essential oil and its delivery means.” is twisted, redundant and folded on itself. Make it easier and clearer! Evaluate if my proposal could be better: “Clarifying the effect of the essential oil of garlic on the reproductive behavior of S. cerealella will be useful not only to understand the specific objective of DATS but will be able to provide appropriate indications on the most suitable formulation of garlic essential oil and its delivery methods.”

Material and Methods

Lines 79-80 – Replace: “The laboratorial S. cerealella colony that feed on wheat, was originally collected from Wuhan, 79 Hubei Province, China” with “The strain of S. cerealella associated to wheat and maintained in laboratory for the experimental tests with DATS, came from Wuhan, Hubei province, China, where it was collected in . . .” Specify when!

Line 80 – Replace: “These moths were kept” with: “This population was kept". How long had the moth population been raised in your laboratory?

Line 81 – Replace: “each kernel” with: “each grain”

Line 82 – Replace: “The 1-day old adults S. cerealella” with: “The 1-day old S. cerealella adults”

Lines 88-89 – Replace: “Ten pairs of 1-day-old virgin adults of S. cerealella were treated with DATS at LC20 (LC20 = 0.015 88 μL/L) by fumigation. The fumigation was conducted in a sealed jar at the rearing conditions.” with: “Set of ten pairs of one male and one female, both 1-day-old virgin adults of S. cerealella, were treated by fumigation with DATS at a dose of 0.015 μL/L (LC20) and compared with sets of as many pairs of identical adults non-fumigated. Transparent glass tubes of about 1000 ml (10 cm in diameter x 125 cm in height), provided with a hermetic screw cap and kept at the rearing conditions, were used for treatments.”

Lines 89-91 – Delete all the sentence: “A filter paper (diameter 2.0 cm) was impregnated with LC20 of DATS and then attached to an undersurface of a 1000 ml glass jar’s (10 cm in diameter ×125 cm in height) screw cap”. Why did you use so long and narrow glass tubes? Moreover note that 5x5x3.14x125=9812! How did you attach the 2.0 cm diameter filter paper diskettes to the inner surface of the glass cylinders?

Lines 91-94 - Replace the three sentences: “The cap was tightly screwed onto the jar, which contained 10 pairs (males and females) of 1-day-old virgin adults. Jars containing the untreated filter paper were considered as controls. Each jar also contained a folded filter paper as an oviposition site for female moths” with: “A filter paper (diameter 2.0 cm), impregnated with LC20 of DATS (treated) or neutral (control) was attached (by what?) to the lower inner surface of the glass tube, together with folded filter paper (of what size?) as a site of egg laying. At the end of the preparation of the device for the tests, each tube contained 20 (10 males and 10 females) virgin 1-day-old adults, filter paper as support for the same adults, and fumigant or neutral element respectively for the treatment and for control”. Did you measure the concentration of DATS in the air present in the tube set up for the test? How much was it? Specify if and what type of liquid food you provided to adults.

Lines 95-96 – With reference to the sentence: “At the same time, all eggs on the folded filter paper were reared at the rearing conditions for two days”, where and how were the glass tubes accommodated and managed? How did you manage and inspect the biological material reared in the glass tubes?

Line 100 – The sentence: “Three replicates of all groups were set up” is not perfectly clear: improve it.

Lines 103 – Replace: “to observe” with: “to ascertain”

Line 107 – Replace: “Jars” with “devices”

Line 111 – Replace: “number of spermatophore” with number of spermatophores”

Lines 112-113 The sentence: “A mated female produced a fine spermatophore in the bursa copulatrix” denotes a serious cognitive deficiency! Spermatophore is not produced by the female but by the male which in the Lepidoptera Dytrisia deposits it in the "bursa copulatrix" at the end of mating.

Lines 113-114 - Replace: “number of spermatophore” with number of spermatophores”

Line 116 – Replace: “into a 1000 ml glass jar’s” with: “into a glass cylinder of about 1000 mL"

Lines 117-121 - Make the description of these “materials and methods” easier and clearer

Line 125 - Rewrite the sentence: “A couple of both sex emergence after 1 day was randomly paired and then mated in a 1 L jar” in order to make the procedure not only intuitable but intelligible.

Lines 129-130 – Replace: “The data were not considered of the dead insects before the 4th day” with: “The data concerning the insects dead before the 4th day were not considered”

Line 123-134 - Rewrite the three sentences in proper English to express the procedures correctly

Line 135 – Replace: “Ma et al. (Ma et al., 2017)” with: “Ma et al. (16)”

Line 141 – Replace: “Also, the covariance analysis was performed” with: “Furthermore, covariance analysis was performed"

Line 142 – Clarify the expression: “the moth’s mating frequency data.”

Results

Line 155 – Replace: “the reduction in eggs deposition didn’t caused by the moth death” with: “the reduction in egg laying was not caused by the death of the moths"

Line 186 – Replace: “the results presented in Table 1 showed” with: “the results presented in Table 1 show”

Line 193 – Replace: “Fig3” with: “Fig. 3”

Lines 195-196 - Rewrite the sentence: “But with DATS treatment, the increased rate of moth’s mating frequency was slower significantly compared with the control” in proper English to better express the result.

Line 201 – Replace: “The calling rhythm of female moths were changed” with: “The calling rhythm of female moths changed.”

Line 229 – Replace: “The daily mean copulation duration ... were observed” with: “The daily mean copulation duration ... was observed”

Discussion

Line 249 – With reference to the expression: “With the advantages of no toxicity to non-target organisms and vertebrates”, what are the scientific bases that allow you to say that plant-based compounds are not toxic to non-target organisms and vertebrates? It is not correct to make these generic and superficial statements in a scientific publication!

Lines 252-253 – Replace: “of the S. cerealella” with: “of S. cerealella

Lines 257-259 - Replace: "DATS and DADS an active component of garlic essential oil, also possessed insecticidal characteristics to inhibit the oviposition of the moth" with: "DATS and DADS showed properties to inhibit the oviposition by the moth". The words “an active component of garlic essential oil” are redundant, and the “inhibition of oviposition” is not an insecticide characteristic!

Lines 276-280 – The two sentences present in these lines are generic and excessive digressions!

Lines 271-273 – Here too there are excessive digressions. In any case it is a good rule to add author and taxonomic position to the Linnaean binomials when cited the first time. Gryllus velitisa ? or Gryllus veletis Alexander & Bigelow, 1960 (Orthoptera: Gryllidae); Gryllus lineaticeps Stål, 1861 (Orthoptera: Gryllidae).

Line 273 – Replace: “Agrotis ypsilon's fecundity” with: “Fecundity of Agrotis ipsilon (Hufnagel, 1766) (Lepidopera: Noctuidae)

Lines 274-275 – Rewrite the sentence: “which in results the eggs deposited by the virgin female were less, the female mated for three times had the most off springs” in understandable English

Lines 283-284 – Replace: “the circadian mating rhythms and calling periodicity were changed significantly” with: “the circadian mating rhythms and calling periodicity changed significantly”

Lines 287-289 - Rewrite the sentence: “It is reported that the sex pheromone titer and PBAN-like activity of the female, Ostrinia furnacalis(Gtrinia fuere reduced after treated with deltamethrin” in understandable English and add author and taxonomic position [Ostrinia furnacalis (Guenée, 1854) (Lepidoptera: Crambidae)]

Lines 292-296 – The three sentences present in these lines are generic and excessive digressions

Line 297 – Replace: “D. melanogaster” with: “Drosophila melanogaster Meigen, 1830 (Diptera: Drosophilidae)

Lines 297-299 - Also the two sentences present in these lines are generic and excessive digressions

Line 303 – Replace: “the moth’s reproductive events” with: “the moth reproductive events”

Lines 306-307 – Replace: “Taken all together, this study suggested that DATS acts as an ovipositional inhibitor against the adult S. cerealella at the sub-lethal concentration by the reduction of the moth’s mating duration” with: “Taken all together, these studies suggest that DATS acts as an ovipositional inhibitor against the S. cerealella adults at the sub-lethal concentration by the reduction of the mating duration”

References

Line 322 - Carefully follow the specific editorial rules. In particular separate an author from the next with a semicolon and use: italic font for binomials, journal name, and issue number; bold round for the publication year; round for authors, titles and pages

Author Response

Respected reviewers,

We are grateful to the reviewers for giving time to review our manuscript. We deeply appreciate the reviewer’s valuable comments and suggestions to improve our manuscript. We tried our best to improve the quality of manuscript in light of reviewer’s comments; meanwhile, the manuscript was polished by a professional scientific editing service (Wiley Editing Services). We hope that the revised manuscript will meet the standards of publication in your good journal.

The point-by-point answers and explanations for all comments were listed below.

Note: Reviewer#2 responses are highlighted with Yellow where possible in the revised manuscript.

Comment 1- Lines 1-5 Replace: “Effect of diallyl trisulfide, an active substance from garlic essential oil, on the reproductive behavior of adults of grain moth, Sitotroga cerealella (Lepidoptera: Gelechiidae)” with: “Effect of diallyl trisulfide on the reproductive behavior of the grain moth Sitotroga cerealella (Lepidoptera: Gelechiidae)”

Response- Thank you for your valuable suggestion, replaced the title as per reviewer’s kind suggestion in the revised manuscript (Line 1-4).

Comment 2- Line 16 - Since Olivier, in 1789, described the Angoumois grain moth as Alucita cerealella, replace "Olivier" with: "(Olivier)" or with: "(Olivier, 1789)

Response- Replace the words as per reviewer’s kind suggestion in the revised manuscript (Line 12).

Comment 3- Lines 17-18 – Replace: “in terms of effective population control 17 of moths,” with: “Replace with: "in terms of effective control of the moth population"

Response- Thank you for your valuable suggestion, we did the revision as your kind suggestion in the revised manuscript (Line 16).

Comment 4- Line 19 – Replace: “at LC20 (LC20 = 0.015μL/L)” with: “at a dose of 0.015 μL/L (LC20).”

Response- As per reviewer’s kind comment, we replaced the statement with a corrected sentence in the revised manuscript (Line 17-18).

Comment 5- Line 20 – Replace: “when treated the moth” with: "when the moth was treated"

Response- As per reviewer’s comment, we replaced “when treated the moth” with "when the moth was treated" in the revised manuscript (Line 19).

Comment 6- Line 21 – Replace: “DATS, also,” with: “DATS. Furthermore,” and: “female’s” with: “female” Line 23 – Replace: “Also,” with: “Moreover,”

Response- As per reviewer’s comment, in the revised manuscript (Line 19), we replaced “DATS, also,” with “DATS. Furthermore,” and: “female’s” with “female”; and in the Line 22, we replaced “Also,” with “Moreover,”.

Comment 7- Line 24 - Replace: “with the existence” with: “in presence”

Response- As per reviewer’s good suggestion, in the revised manuscript (Line 22-23), we replaced “with the existence” with “in presence”.

Comments 8- Line 27 – Replace: “Additionally,” with: “In addition,”

Response- As per reviewer’s good suggestion, in the revised manuscript (Line 25), we replaced “Additionally,” with “In addition,”.

Comments 9- Lines 31-32 – The sentence “an efficient alternative to synthetic insecticides for the control of pest without harming non-target organisms and ecosystem” is a pretty bombastic phrase, without scientific support. Which data demonstrate the selectivity of DATS towards no target and beneficial insects?

 Response- Thank you so much for pointing out and nice suggestion. We are very sorry to wrongly present, we agree with the reviewer suggestion and made changes in the sentence (without harming non-target organisms and ecosystem) with (low toxicity to non-target organisms and ecosystem) from manuscript Line 29-30. The sentence was changed with the support of published data which states that botanicals or garlic oil and its principle substances possess minimum toxicity to environment and non-target organisms. The references are:

Ramakrishnan, V., Chintalwar, G.J., Banerji, A.. 1989. Environmental persistence of diallyl disulfide an insecticidal principle of garlic and its metabolism in mosquito, Culex pipiens quinquifasciatus Say. Chemosphere 18:1525-1529. Saeed, R.M.A., Zayed, A.A., El Namaky, A.H., Ismail, H.M., Mady, H.Y.. 2010. Biochemical studies on Culex pipiens (L.) (Diptera: Culicidae) exposed to Allium sativum, Citrus limon and Bacillus thuringiensis israelensis with reference to assessment of the biosafety on albino mice. Global Veterinaria 4(1): 22-33. Isman, M.B.. 2006. Botanical insecticides, deterrents, and repellents in modern agriculture and an increasingly regulated world. Annual Review of Entomology. 51: 45-66. Zhao, N.N., Zhang, H., Zhang, X.C., Luan, X, B., Zhou, C., Liu, Q.Z., Shi, W. P., Liu, Z. L. 2013. Evaluation of acute toxicity of essential oil of garlic (Allium sativum) and its selected major constituent compounds against overwintering Cacopsylla chinensis (Hemiptera: Psyllidae). Journal of Economic Entomology, 106(3):1349-1354..

Comments 10- Lines 33-34 - Taking into account that the first three keywords are already present in the title and that the editorial rules provide for an alphabetical order, replace: “Diallyl trisulfide; Sitotroga cerealella; reproductive behavior; circadian mating rhythms; courtship periodicity; inhibiting oviposition” with: "botanicals; circadian mating rhythms; courtship periodicity; garlic essential oil; inhibiting oviposition; pest management; stored product protection”

Response- In the revised manuscript (Line 31-32), we replaced the key words according to the reviewer’s kind suggestion.

Comments 10- Line 37 – Replace: “Sitotroga cerealella” with: “Sitotroga cerealella (Olivier) (Lepidopera: Gelechiidae)”. This formulation cannot be present in "Abstract" and not in the “Introduction”

Response- Thank you so much for your valuable suggestion, we replaced “Sitotroga cerealella” with: “Sitotroga cerealella (Olivier) (Lepidoptera: Gelechiidae)” in Line 35.

Comments 11- Line 39 – Delete: “insect” and “systems” because they are redundant

Response- As per reviewer’s kind comment, we deleted words “insect” and “systems” in the revised manuscript.

Comments 12- Line 41 – Replace: “the applications of synthetic insecticide” with: “the applications of synthetic insecticides”

Response- As per reviewer’s good suggestion, we replaced “the applications of synthetic insecticide” with “the applications of synthetic insecticides”( Line 39-40).

Comments 13- Lines 41-42 – Replace: “alternatives strategies to conventional insecticides” with: "alternative methods to those based on conventional pesticides"

Response- Thank you so much for valuable suggestion, we replaced the “alternatives strategies to conventional insecticides” with "alternative methods to those based on conventional pesticides" in the revised manuscript (Line 40-41).

Comments 14- Line 48 – Replace: “Mostly female moths attract the male” with: “Mostly female moths attract males”

Response- As per reviewer’s nice suggestion, we replaced “Mostly female moths attract the male” with “Mostly female moths attract males” in the revised manuscript (Line 126).

Comments 15- Line 50 – Replace: “After the mate recognition, the male displayed” with: “After the mate recognition, the male displays”

Response- As per reviewer’s good suggestion, we replaced “After the mate recognition, the male displayed” with “After the mate recognition, the male displays” in the revised manuscript (Line 128).

Comments 16- Line 51 – Replace: “and copulated” with: “and copulates”

Response- We replaced “and copulated” with “and copulates” in the revised manuscript (Line 129) as per reviewer’s suggestion.

Comments 17- Line 52 – Replace: “a large number of moth species mated” with: " a large number of moth species mate"

Response- We replaced “a large number of moth species mated” with " a large number of moth species mate" as per reviewer’s suggestion in the revised manuscript (Line 130).

Comments 18- Line 56 – Replace: “of the S. cerealella which feed” with: "of S. cerealella which feeds"

Response- We replaced “of the S. cerealella which feed” with "of S. cerealella which feeds" as per reviewer’s suggestion in the revised manuscript (Line 134).

Comments 19- Line 60 – Replace: “plant resource materials . . . possessed the feasibility” with: “plant resource materials . . . hold the feasibility”

Response- As per reviewer’s good suggestion, we replaced “plant resource materials . . . possessed the feasibility” with “plant resource materials . . . hold the feasibility” in the revised manuscript (Line 137-138).

Comments 20- Line 60 – Replace: “the control of stored-product insect pests” with: “the control of stored-product pests”. The term “insect” is redundant!

Response- As per reviewer’s good comment, we replaced “the control of stored-product insect pests” with “the control of stored-product pests” in the revised manuscript (Line 138).

Comments 21- Line 63 – Replace: “female S. cerealella” with: “S. cerealella female”

Response- As per reviewer’s kind suggestion, we replaced “female S. cerealella” with “S. cerealella female” in the revised manuscript (Line 140-141).

Comments 22- Line 64 – Replace: “including the effects on mate-locating” with: “including those on mate-locating”

Response- As per reviewer’s kind suggestion, we replaced “including the effects on mate-locating” with “including those on mate-locating” in the revised manuscript (Line 141-142).

Comments 23- Line 65 – Replace: “While our previous molecular level studies” with: “While our previous molecular studies”

Response- As per reviewer’s kind suggestion, we replaced “While our previous molecular level studies” with “While our previous molecular studies” in the revised manuscript (Line 142-143).

Comments 24- Line 67 – Replace: “the real reasons how major components” with: “the real reasons why major components”

Response- As per reviewer’s good suggestion, we replaced “the real reasons how major components” with “the real reasons why major components” in the revised manuscript (Line 144-145).

Comments 25- Line 68 – Replace: “female’s fecundity” with: “female fecundity”

Response- As per reviewer’s good suggestion, we replaced “female’s fecundity” with “female fecundity” in the revised manuscript (Line 146).

Comments 26- Line 70 – Replace: “observe the mating behavior” with: “detect anomalies in mating behavior”

Response- As per reviewer’s good suggestion, we replaced observe the mating behavior” with “detect anomalies in mating behavior” in the revised manuscript (Line 148).

Comments 27- Lines 73-76 – The sentence: “To elucidate the effect of garlic essential oil on the reproductive behavior of S. cerealella is of practical importance for insect control because it will not only help to clarify the specific target of DATS but also provide useful information on the most suitable formulation of garlic essential oil and its delivery means.” is twisted, redundant and folded on itself. Make it easier and clearer! Evaluate if my proposal could be better: “Clarifying the effect of the essential oil of garlic on the reproductive behavior of S. cerealella will be useful not only to understand the specific objective of DATS but will be able to provide appropriate indications on the most suitable formulation of garlic essential oil and its delivery methods.”

Response- Thank you so much for your kind suggestion and we agree with reviewer suggestion and changed the sentence according to reviewer expert opinion (Line 151-154).

Comments 28- Lines 79-80 – Replace: “The laboratorial S. cerealella colony that feed on wheat, was originally collected from Wuhan, 79 Hubei Province, China” with “The strain of S. cerealella associated to wheat and maintained in laboratory for the experimental tests with DATS, came from Wuhan, Hubei province, China, where it was collected in . . .” Specify when!

Response- Thank you so much for your expert suggestion, the sentence was changed and information was added according to reviewer suggestion (Line 157-164).

Comments 29- Line 80 – Replace: “These moths were kept” with: “This population was kept". How long had the moth population been raised in your laboratory?

Response- Thank you so much for your kind comments, the sentence was replaced according to the reviewer kind suggestion (Line 158-159), and in our laboratory, we are working on this insect from 2012 and we continually raising the population, for which every week the grains are inspected and the jars are dated.

Comments 30- Line 81 – Replace: “each kernel” with: “each grain”

Response- We replaced “each kernel” with “each grain” according to the reviewer suggestion in the revised manuscript (Line 161).

Comments 31- Line 82 – Replace: “The 1-day old adults S. cerealella” with: “The 1-day old S. cerealella adults”

Response- We replaced the sentence “The 1-day old adults S. cerealella” with “The 1-day old S. cerealella adults” according to the reviewer expert comment (Line 164).

Comments 32- Lines 88-89 – Replace: “Ten pairs of 1-day-old virgin adults of S. cerealella were treated with DATS at LC20 (LC20 = 0.015 88 μL/L) by fumigation. The fumigation was conducted in a sealed jar at the rearing conditions.” with: “Set of ten pairs of one male and one female, both 1-day-old virgin adults of S. cerealella, were treated by fumigation with DATS at a dose of 0.015 μL/L (LC20) and compared with sets of as many pairs of identical adults non-fumigated. Transparent glass tubes of about 1000 ml (10 cm in diameter x 125 cm in height), provided with a hermetic screw cap and kept at the rearing conditions, were used for treatments.”

Response- Thank you so much for your expert opinion, we replaced the sentence according to your suggestion (Line 284-288).

Comments 33- Lines 89-91 – Delete all the sentence: “A filter paper (diameter 2.0 cm) was impregnated with LC20 of DATS and then attached to an undersurface of a 1000 ml glass jar’s (10 cm in diameter ×125 cm in height) screw cap”. Why did you use so long and narrow glass tubes? Moreover note that 5x5x3.14x125=9812! How did you attach the 2.0 cm diameter filter paper diskettes to the inner surface of the glass cylinders?

Response- Thank you so much for pointing out our mistake, we deleted the sentence and modified and replaced according to your expert suggestions (comment 33) in revised manuscript (Line 288-293).

In response to your kind comments, we regret that we wrongly described and changed the word here and elsewhere in whole manuscript, we used (1000 ml, 10 cm diameter × 13 cm height) not (10 cm in diameter ×125 cm in height), same as used by another researcher of our laboratory (Yang et al. 2012 and Min ma et al. 2017).

Yang, F.L., Zhu, F., Lei, C.L.. Insecticidal activities of garlic substances against adults of grain moth, Sitotroga cerealella (Lepidoptera: Gelechiidae). Insect Science 2012, 19, 205-212.

 Ma, M., Ying, Y.Y., He, Z.Y., Lu, Y., Li, X.G., Lei, C.L., Yang F.L..Comparative transcriptome analysis reveals potential gene targets for ovipositional inhibition by diallyl trisulfide against female Sitotroga cerealella. Agri Gene 2017: 12-18.

Comments 33- Lines 91-94 - Replace the three sentences: “The cap was tightly screwed onto the jar, which contained 10 pairs (males and females) of 1-day-old virgin adults. Jars containing the untreated filter paper were considered as controls. Each jar also contained a folded filter paper as an oviposition site for female moths” with: “A filter paper (diameter 2.0 cm), impregnated with LC20 of DATS (treated) or neutral (control) was attached (by what?) to the lower inner surface of the glass tube, together with folded filter paper (of what size?) as a site of egg laying. At the end of the preparation of the device for the tests, each tube contained 20 (10 males and 10 females) virgin 1-day-old adults, filter paper as support for the same adults, and fumigant or neutral element respectively for the treatment and for control”. Did you measure the concentration of DATS in the air present in the tube set up for the test? How much was it? Specify if and what type of liquid food you provided to adults.

Response- Thanks for your good suggestion, we replaced the three sentences and added information according to reviewer’s suggestions in the revised manuscript (Line 288-293). 

Thanks for your expert comment. Yes, we measured the concentration of DATS in the air present in the tube set up for test and it was 0.015µL/L, the dilution as our dose LC20 means 0.015µL per liter of air. The treated method was as following: 1.5µL DATS was dissolved into 985µL acetone by a micropipette, 10µL of the solution was picked up and deposited on a filter paper (diameter 2.0 cm). The filter paper was air dried for 2 minutes to evaporate the acetone. And then the filter paper (diameter 2.0 cm), impregnated with LC20 of DATS (treated) was attached by gum to the inner surface of the lid of a glass tube (1000 ml, 10 cm in diameter x 13 cm in height). The lid was put onto a glass tube for fumigation. Concentration for fumigation was 0.015 μL/L air. The same method was followed by Huang et al.(2000) and Yang et al. (Yang et al. 2012).

During the experiments, considering there will be adverse side effect of acetone on adult moths, so we evaluated and confirmed that there was no difference between the groups treated with 10µL acetone alone and the blank controls without any treatment for the adults’ No. eggs laid, mortality rate, and mating frequency within 24h. Please see the following figure:

Fig.1 effects of acetone on the total number eggs laid, mortality rate and mating frequency of adult moths 48 hrs after treatment

Note: CK means groups without any treatment/ blank controls and BT means acetone-treated groups in Fig.1.

We did the experiment on the adult moths which don’t need any food, and we did the experiment in complete control condition. The glass tubes were packed with cotton cloth having pores and placed in incubator with 28±1°C, 75%±5% RH with a photoperiod of 14:10 (L:D) h.

Huang, Y., Chen, S.X., Ho, S.H.. Bioactivities of methyl allyldisulfide and diallyl trisulfide from essential oil of garlic to two species of stored-product pests, Sitophilus zeamais (Coleoptera: Curculionidae) and Tribolium castaneum (Coleoptera: Tenebrionidae). Journal Economic Entomology 2000, 93(2):537–543.

Yang, F.L., Zhu, F., Lei, C.L.. Insecticidal activities of garlic substances against adults of grain moth, Sitotroga cerealella (Lepidoptera: Gelechiidae). Insect Science 2012, 19: 205-212.

Comments 34- Lines 95-96 – With reference to the sentence: “At the same time, all eggs on the folded filter paper were reared at the rearing conditions for two days”, where and how were the glass tubes accommodated and managed? How did you manage and inspect the biological material reared in the glass tubes?

Response- We did the experiment in complete control condition. The glass tube was packed with a lid and placed in incubator with 28±1°C, 75%±5% RH with a photoperiod of 14:10 (L:D) h. during experiment period, the glass tubes are transparent and easily saw the insect’s activity which were placed in the incubator and inspect the insects after every 3 to 4 hours interval daily. Furthermore, all rearing condition were made ideal for eggs and insects.

Comments 35- Line 100 – The sentence: “Three replicates of all groups were set up” is not perfectly clear: improve it.

Response- Thanks for your good comment, we changed the sentence in the revised manuscript (Line 299).

Comments 36- Lines 103 – Replace: “to observe” with: “to ascertain”

Response- We replaced the “to observe” with “to ascertain” according to reviewer’s suggestion in the revised manuscript (Line 303).

Comments 37- Line 107 – Replace: “Jars” with “devices”

Response- We replaced “Jars” with “devices” according to reviewer’s suggestion in the revised manuscript (Line 307).

Comments 38- Line 111 – Replace: “number of spermatophore” with number of spermatophores”

Response- We replaced “number of spermatophore” with “number of spermatophores” according to reviewer’s suggestion in the revised manuscript (Line 311).

Comments 39- Lines 112-113 The sentence: “A mated female produced a fine spermatophore in the bursa copulatrix” denotes a serious cognitive deficiency! Spermatophore is not produced by the female but by the male which in the Lepidoptera Dytrisia deposits it in the "bursa copulatrix" at the end of mating.

Response- Thank you so much for identifying our mistake, the sentence changed in the revised manuscript (Line 312) according to reviewer’s expert suggestion.

Comments 40- Line 113-114 – Replace: “number of spermatophore” with number of spermatophores”

Response- We replaced “number of spermatophore” with “number of spermatophores” according to reviewer’s suggestion in the revised manuscript (Line 311).

Comments 41- Line 116 – Replace: “into a 1000 ml glass jar’s” with: “into a glass cylinder of about 1000 mL"

Response- We replaced “into a 1000 ml glass jar’s” with “into a glass cylinder of about 1000 mL" according to reviewer’s suggestion in the revised manuscript (Line 316).

Comments 42- Lines 117-121 - Make the description of these “materials and methods” easier and clearer

Response- We changed the description in the revised manuscript (Line 315-323) according to reviewer expert suggestion.

Comments 43- Line 125 - Rewrite the sentence: “A couple of both sex emergence after 1 day was randomly paired and then mated in a 1 L jar” in order to make the procedure not only intuitable but intelligible.

Response- we rewrite the sentence according to reviewer’s kind suggestion in the revised manuscript (Line 325).

Comments 44- Lines 129-130 – Replace: “The data were not considered of the dead insects before the 4th day” with: “The data concerning the insects dead before the 4th day were not considered”

Response- Thank you so much for your suggestion, as per your below suggestion (comment 45), the sentence was modified by English editing company in the revised manuscript (Line 329).

Comments 45- Line 123-134 - Rewrite the three sentences in proper English to express the procedures correctly

Response- The whole manuscript was edited by English editing company (Wiley Editing Services) including these three sentences in the revised manuscript (Line316-329).

Comments 46- Line 135 – Replace: “Ma et al. (Ma et al., 2017)” with: “Ma et al. (16)”

Response- We replaced according to reviewer’s kind suggestion in the revised manuscript (Line 473).

Comments 47- Line 141 – Replace: “Also, the covariance analysis was performed” with: “Furthermore, covariance analysis was performed"

Response- We replaced the sentence according to reviewer’s good suggestion in the revised manuscript (Line 487-488).

Comments 48- Line 142 – Clarify the expression: “the moth’s mating frequency data.”

Response- Thank you so much for your kind comment, we changed the description of the (Statistical analysis) in the revised manuscript (Line 481-490).

Comments 49- Line 155 – Replace: “the reduction in eggs deposition didn’t caused by the moth death” with: “the reduction in egg laying was not caused by the death of the moths"

Response- Thank you so much for your kind comment, we replaced the sentence in the revised manuscript (Line 498-499).

Comment 50- Line 186 – Replace: “the results presented in Table 1 showed” with: “the results presented in Table 1 show”

Response- Thank you so much for your kind comment, we modified the description of the whole paragraph as instructions of English editing company and replaced the above word as (Table 1) in the revised manuscript (Line 534-540).

Comment 51- Line 193 – Replace: “Fig3” with: “Fig. 3

 Response- Thank you so much for your kind comment, we replaced the sentence in the revised manuscript (Line 544).

Comment 52- Lines 195-196 - Rewrite the sentence: “But with DATS treatment, the increased rate of moth’s mating frequency was slower significantly compared with the control” in proper English to better express the result.

Response- Thank you so much for your suggestion, the sentence was rewritten in better English with the help of English editing company (Line 545-546).

Comment 53- Line 201 – Replace: “The calling rhythm of female moths were changed” with: “The calling rhythm of female moths changed.”

Response- Thank you so much for your kind comment, we replaced the sentence in the revised manuscript (Line 554).

Comments 54- Line 229 – Replace: “The daily mean copulation duration ... were observed” with: “The daily mean copulation duration ... was observed”

Response- Thank you so much for your kind comment, we replaced the sentence in the revised manuscript (Line 581).

Comment 55- Line 249 – With reference to the expression: “With the advantages of no toxicity to non-target organisms and vertebrates”, what are the scientific bases that allow you to say that plant-based compounds are not toxic to non-target organisms and vertebrates? It is not correct to make these generic and superficial statements in a scientific publication!

Response- Thank you so much for pointing out and nice suggestion. We are very sorry for wrongly presented, we agree with the reviewer suggestion and made changes in the sentence in the revised manuscript (Line 607). The sentence was changed with the support of published data and added the references.

Comments 56- Lines 252-253 – Replace: “of the S. cerealella” with: “of S. cerealella”

Response- Thank you so much for your kind comment, we replaced the sentence in the revised manuscript (Line 610).

Comments 57- Lines 257-259 - Replace: "DATS and DADS an active component of garlic essential oil, also possessed insecticidal characteristics to inhibit the oviposition of the moth" with: "DATS and DADS showed properties to inhibit the oviposition by the moth". The words “an active component of garlic essential oil” are redundant, and the “inhibition of oviposition” is not an insecticide characteristic!

Response- Thank you so much for your kind comment, we replaced the sentence in the revised manuscript (Line 628).

Comments 58- Lines 276-280 – The two sentences present in these lines are generic and excessive digressions!

Response- Thank you so much for your suggestion and also an advice from another reviewer, we deleted these two sentences from discussion.

Comments 59- Lines 271-273 – Here too there are excessive digressions. In any case it is a good rule to add author and taxonomic position to the Linnaean binomials when cited the first time. Gryllus velitisa? or Gryllus veletis Alexander & Bigelow, 1960 (Orthoptera: Gryllidae); Gryllus lineaticeps Stål, 1861 (Orthoptera: Gryllidae).

Response- Thank you so much for your suggestion, we changed the taxonomic position and citation according to your kind suggestion in the revised manuscript (Line 636-638).

Comments 60- Line 273 – Replace: “Agrotis ypsilon's fecundity” with: “Fecundity of Agrotis ipsilon (Hufnagel, 1766) (Lepidopera: Noctuidae)

Response- Thank you so much for your suggestion we replaced the words in the revised manuscript (Line 638-639).

Comments 61- Lines 274-275 – Rewrite the sentence: “which in results the eggs deposited by the virgin female were less, the female mated for three times had the most off springs” in understandable English

Response- Thank you so much for your suggestion, we changed the description of the sentence in the revised manuscript (Line 639-640).

Comments 62- Lines 283-284 – Replace: “the circadian mating rhythms and calling periodicity were changed significantly” with: “the circadian mating rhythms and calling periodicity changed significantly”

Response- Thank you so much for your suggestion, we replaced the sentence in the revised manuscript (Line 648).

Comments 63- Lines 287-289 - Rewrite the sentence: “It is reported that the sex pheromone titer and PBAN-like activity of the female, Ostrinia furnacalis(Gtrinia fuere reduced after treated with deltamethrin” in understandable English and add author and taxonomic position [Ostrinia furnacalis (Guenée, 1854) (Lepidoptera: Crambidae)]

Response- Thank you so much for your suggestion however, we got advice from another reviewer, so we deleted these sentences from discussion.

Comments 64- Lines 292-296 – The three sentences present in these lines are generic and excessive digressions

Response- Thank you so much for your suggestion however, we got advice from another reviewer, so we deleted these sentences from discussion.

Comments 65- Line 297 – Replace: “D. melanogaster” with: “Drosophila melanogaster Meigen, 1830 (Diptera: Drosophilidae)

Response- Thank you so much for your suggestion however, we have advice from another reviewer, so we deleted these sentences from discussion.

Comments 66- Lines 297-299 - Also the two sentences present in these lines are generic and excessive digressions

Response- Thank you so much for your suggestion however, we got advice from another reviewer, so we deleted these sentences from discussion.

Comments 67- Line 303 – Replace: “the moth’s reproductive events” with: “the moth reproductive events”

Response- Thank you so much for your suggestion however, we got advice from another reviewer, so we deleted these sentences from discussion.

Comments 68- Lines 306-307 – Replace: “Taken all together, this study suggested that DATS acts as an ovipositional inhibitor against the adult S. cerealella at the sub-lethal concentration by the reduction of the moth’s mating duration” with: “Taken all together, these studies suggest that DATS acts as an ovipositional inhibitor against the S. cerealella adults at the sub-lethal concentration by the reduction of the mating duration”

Response- Thank you so much for your suggestion, we replaced the sentence in the revised manuscript (Line 651-652).

Comments 69- Line 322 - Carefully follow the specific editorial rules. In particular separate an author from the next with a semicolon and use: italic font for binomials, journal name, and issue number; bold round for the publication year; round for authors, titles and pages

Response- Thank you so much for your nice suggestion, all the references were formatted again according to the journal specific rules.

With the incorporation of constructive suggestions from reviewers, the manuscript has been clearly improved. We hope that the revised manuscript will satisfy the reviewers and that you will find the paper is now suitable for publication. Thanks again for your timely consideration of our manuscript.

Dr Feng-Lian Yang

Hubei Insect Resources Utilization and Sustainable Pest Management Key Laboratory

Huazhong Agricultural University

Wuhan 430070, People’s Republic of China

Tel./Fax: +86-27-87287207

E-mail: yangfenglian@mail.hzau.edu.cn

Round 2

Reviewer 1 Report

The authors have made a good faith effort to address most of the points raised in my first review.  There are a couple of minor points that should still be addressed in the interests of clarity:

L 18 and elsewhere throughout the manuscript, state explicitly that the dose is 0.015 uL/L in air. L 22, according to the text, the delay in mating was only significant on days 2 and 3, not on all 4 days. L 27 and throughout, replace volume of pheromone with amount of pheromone. L 89 on, explicitly and completely describe how the chemical was applied, i.e., 10 ul of a solution that was 1.5 microliters per L in acetone was applied to a filter paper and the filter paper was glued... L 120, insert "calling" between "of" and "S." Same L 207 L 116, move "after fumigation" to the end of the sentence L 139-144, the quantification of the pheromone is still not described in enough detail to be able to reproduce the experiment.  I.e., how much hexane was used to extract the glands, was an internal standard used, and what was the injection volume? L 259, at the least, the authors need citations to support their highly debatable statement that there is slow evolution of resistance to plant-based compounds.  L 287 on, this is not a good comparison and I would suggest removing it, i.e., comparing the number of eggs laid by unmated females with females mated three times.  If the authors have a citation comparing females mated a single time with females mated multiple times, that would be much better.

Author Response

Respected editor and reviewer,

We are grateful to the reviewer for giving time to review our manuscript. We deeply appreciate the reviewer’s valuable comments and suggestions to improve our manuscript. We tried our best to improve the quality of manuscript in light of reviewer’s comments. We hope that the revised manuscript will meet the standards of publication in your good journal.

The point-by-point answers and explanations for all comments were listed below.

Note. Reviewer#1 responses are highlighted with light blue where possible in the revised manuscript.

Comment 1- L 18 and elsewhere throughout the manuscript, state explicitly that the dose is 0.015 uL/L in air.

Response- We are thankful to the reviewer kind suggestion and accordingly we state in L 18 and L 90 in revised Manuscript.

Comment 2- L 22, according to the text, the delay in mating was only significant on days 2 and 3, not on all 4 days.

Response- Thanks to the reviewer’s expert comment, in revised Manuscript L 22, we write the information according to the expert suggestion of reviewer.

Comment 3- L 27 and throughout, replace volume of pheromone with amount of pheromone.

Response- Thanks to the reviewer’s expert suggestion, we replaced volume of pheromone with amount of pheromone in the revised manuscript (L 27, 71, 142, 151, 159, 261, 262, 265 and 268).

Comment 4- L 89 on, explicitly and completely describe how the chemical was applied, i.e., 10 ul of a solution that was 1.5 microliters per L in acetone was applied to a filter paper and the filter paper was glued...

Response- We are thankful for reviewer’s expert suggestion and accordingly we added the information in the revised manuscript (L 90 to L 95).

Comment 5- L 120, insert "calling" between "of" and "S." Same L 207

Response- We are thankful for reviewer’s expert suggestion and accordingly we modify the information in revised manuscript (L 126 and L 214).

Comment 6- L 116, move "after fumigation" to the end of the sentence

Response- We are thankful for reviewer’s expert suggestion and accordingly we move the information in the revised manuscript (L 123).

Comment 7- L 139-144, the quantification of the pheromone is still not described in enough detail to be able to reproduce the experiment.  I.e., how much hexane was used to extract the glands, was an internal standard used, and what was the injection volume?

Response- We are thankful for reviewer’s expert suggestion and accordingly we added more details to address the experiment in the revised manuscript (Line 146-152). We didn’t use an internal standard when compared with the amount of pheromone in different treated groups. We just compared the area of sex pheromone extract at a retention time of 12.4 min for different treatments.

Comment 8- L 259, at the least, the authors need citations to support their highly debatable statement that there is slow evolution of resistance to plant-based compounds.

Response- We are thankful for reviewer’s expert suggestion and accordingly we put the reference in the revised manuscript (L 273). In the reference paper in Abstract L 6 and in introduction L 13, the author quotes the statement which supports our statement. The paper is here for reviewer’s kind consideration and also put in revised Manuscript.

Borges, L. M. F.; Sousa, L.A.D.; Barbosa, C.S. Perspectives for the use of plant extracts to control the cattle tick Rhipicephalus (Boophilus) microplus.  Rev. Bras. Parasitol. Vet., 2011( 20): 89-96.

Comment 9- L 287 on, this is not a good comparison and I would suggest removing it, i.e., comparing the number of eggs laid by unmated females with females mated three times.  If the authors have a citation comparing females mated a single time with females mated multiple times, that would be much better.

Response- We are thankful for reviewer’s expert suggestion and accordingly we deleted the statement from the revised manuscript (L 302).

With the incorporation of constructive suggestions from reviewers, the manuscript has been clearly improved. We hope that the revised manuscript will satisfy the reviewers and that you will find the paper is now suitable for publication. Thanks again for your timely consideration of our manuscript.

Dr Feng-Lian Yang

Hubei Insect Resources Utilization and Sustainable Pest Management Key Laboratory

Huazhong Agricultural University

Wuhan 430070, People’s Republic of China

Tel./Fax: +86-27-87287207

E-mail: yangfenglian@mail.hzau.edu.cn
